# Zonal variability of methane trends derived from satellite data

Jonas Hachmeister[1], Oliver Schneising[1], Michael Buchwitz[1], John P. Burrows[1], Justus Notholt[1], and Matthias Buschmann[1]

[1]Institute of Environmental Physics (IUP), University of Bremen FB1, Bremen, Germany

**Correspondence:** J. Hachmeister (jonas_h@iup.physik.uni-bremen.de)

**Abstract.** The Tropospheric Monitoring Instrument (TROPOMI) on-board the satellite Sentinel-5 Precursor (S5P) is part of the latest generation of trace gas monitoring satellites and provides a new level of spatio-temporal information with daily global coverage, which enable the calculation of daily globally averaged $CH_4$ concentrations. To investigate changes of atmospheric methane, the background $CH_4$ level (i.e. the $CH_4$ concentration without seasonal and short-term variations) has to be deter-

mined. $CH_4$ growth rates vary in a complex manner and high-latitude zonal averages may have gaps in the time series, thus simple fitting methods don't produce reliable results. In this manuscript we present an approach based on fitting an ensemble of Dynamic Linear Models (DLMs) to TROPOMI data, from which the best model is chosen with the help of cross-validation to prevent overfitting. This method is computationally fast and not dependent additional inputs, allowing for the fast and continuous analysis of the most recent time series data. We present results of global annual methane increases (AMIs) for the first

4.5 years of S5P/TROPOMI data which show good agreement with AMIs from other sources. Additionally, we investigated what information can be derived from zonal bands. Due to the fast meridional mixing within hemispheres we use zonal growth rates instead of AMIs, since they provide a higher temporal resolution. Clear differences can be observed between Northern and Southern Hemisphere growth rates, especially during 2019 and 2022. The growth rates show similar patterns within the hemispheres and show no short-term variations during the years, indicating that air masses within a hemisphere are well-mixed

during a year. Additionally, the growth rates derived from S5P/TROPOMI data are largely consistent with growth rates derived from CAMS global inversion-optimized (CAMS/INV) data which uses surface observations. In 2019 a reduction in growth rates can be observed for the Southern Hemisphere, while growth rates in the Northern Hemisphere stay stable or increase. During 2020 a strong increase in Southern Hemisphere growth rates can be observed, which is in accordance with recently reported increases in Southern Hemisphere wetland emissions. In 2022 the reduction of the global AMI can be attributed to

decreased growth rates in the Northern Hemisphere, while growth rates in the Southern Hemisphere remain high. Investigations of fluxes from CAMS/INV data support these observations and suggest that the Northern Hemisphere decrease is mainly due to the decrease in anthropogenic fluxes while in the Southern Hemisphere wetland fluxes continued to rise. While the continued increase of Southern Hemisphere Wetland fluxes agrees with existing studies about the causes of observed methane trends, the difference between Northern and Southern Hemisphere methane increases in 2022, hasn't been discussed before and calls for

further research.

# 1 Introduction

Methane ($CH_4$) is one of the most important drivers of climate change with an effective radiative forcing of $1.19$ $Wm^{-2}$ (Arias et al., 2021) and an atmospheric lifetime of 9.1 years (Szopa et al., 2021). The short lifespan of $CH_4$ compared to other greenhouse gases and the large fraction of anthropogenic emissions, makes $CH_4$ emission reduction an attractive strategy to slow down or possibly reduce man-made climate change in the short- to midterm. Accurate knowledge of the atmospheric $CH_4$ concentrations and dry column mixing ratios are therefore essential to improve our knowledge of the sources and sinks of $CH_4$ for science and international environmental policy. The globally averaged surface concentration of $CH_4$ has increased by 156% between 1750 and 2019 reaching $1866 \pm 3.3$ ppb in 2019 (Gulev et al., 2021) and $1917.11$ ppb in June 2023 (Lan et al., 2023).

While the concentrations have risen in total, the trend, i.e. the rate of change in the background level without seasonal or short-term variations, has evolved non-linearly.Global methane concentrations have been observed to increase in the period from the 1980s to 2000 and from 2007 until the present. However, a plateau between 2000-2007 was observed. This is referred to as "stabilization". Whether to define the stabilization period or the period of renewed growth (2007-present) as anomalous has been the subject of debate. There have been a variety of explanations for the observed behavior in the literature (Turner et al., 2019). Recent publications suggest that the period of renewed growth can be attributed to the rise in microbial emissions (Lan et al., 2021; Basu et al., 2022) and that tropical methane emissions explain a majority of recent changes in the atmospheric methane growth rate (Feng et al., 2022). In 2020 and 2021 record methane increases were observed by the Global Monitoring Laboratory of the 'National Oceanic and Atmospheric Administration (NOAA-GML) (Lan et al., 2023) and the Copernicus Climate Change Service (C3S) (c3s, 2023a). The reasons for these increases are still debated, with studies attributing them to increases in wetland emissions and changes in the atmospheric methane sink to varying degrees. The main sink of methane is through reaction with the hydroxyl radical (OH) in the troposphere. The rate of this reaction depends on the concentration of OH, which is determined by its photochemical sources and sinks. Recent studies suggest that the steep decline of nitrogen dioxide ($NO_2$) (Cooper et al., 2022), carbon monoxide (CO) and non-methane volatile organic compound emissions as a result of the measures introduced to control and limit the spread of the COVID-19 pandemic, lowered the levels of OH, and thus led to part of the increase in $CH_4$ concentrations in 2020 and 2021 (Stevenson et al., 2022; Laughner et al., 2021; Peng et al., 2022; Qu et al., 2022; Feng et al., 2023). Additionally, enhanced wetland emissions, especially from tropical wetlands, contributed to the record increases of atmospheric $CH_4$ in 2020/21 (Peng et al., 2022; Feng et al., 2023, 2022; Qu et al., 2022).

The Arctic contains large amounts of soil organic carbon (SOC) which is stored in the permafrost regions (ca. 1300 Pg) of which roughly 800 Pg is perennially frozen (Hugelius et al., 2014). The comparatively high temperature increase in the Arctic, compared to the rest of the world, also called "Arctic amplification" (Serreze and Barry, 2011; Wendisch et al., 2017) may lead to increased permafrost degradation and rapid SOC loss (Plaza et al., 2019) by the release of carbon dioxide ($CO_2$) and/or methane. Latitudinally resolved growth rates are especially interesting in this regard and provided the initial motivation for this

study.

In this paper we present methane growth rates and annual methane increases (AMIs) derived from Sentinel-5P/TROPOMI XCH$_4$ data using a Dynamic Linear Model (DLM) approach. In the second section we present the data used. Next, we describe our method for calculating these growth rates, which is divided into four parts: (i) we discuss the preparation of the data; (ii) we provide a brief introduction into DLMs; (iii) we discuss our ensemble approach which utilizes cross validation to find the

65 optimal DLM configuration for a given time series; (iv) we provide a method to calculate a bias related to the satellite sampling. In the fourth section we present global annual methane increases (AMIs) for the first 4.5 years of S5P/TROPOMI data and compare these to AMIs from other sources. In the fifth section we investigate zonal growth rates derived from 20° latitudinal bands to provide spatial information to the global AMIs. Additionally, we compare the growth rates to growth rates derived from CAMS global inversion-optimized methane data (CAMS/INV) which assimilate either surface observations or surface

and satellite data. In the sixth section we investigate CAMS/INV fluxes to help with the interpretation of our previous results. Finally, we summarize our results and discuss potential future uses of this method and suggestions for further research. In the Appendix we provide additional information about our method and further results which are not included in the main text.

## 2 Data

### 2.1 Sentinel-5P/TROPOMI WFMD product

The Sentinel-5 Precursor (S5P) satellite was launched on 13 October 2017 and has since delivered high quality data from its only scientific instrument, TROPOMI, which is a nadir viewing passive grating imaging spectrometer. Combined with a near-polar, sun-synchronous orbit, the swath width of 2600 km provides daily global coverage. Due to the orbit geometry and swath overlap multiple observations per day are possible in the polar regions. The spatial resolution depends on the bands and is $5.5 \times 7$ km$^2$ for the short-wave infrared (SWIR) band ($7 \times 7$ km$^2$ before August 2019) (Ludewig, 2021). Methane is

retrieved from TROPOMI measurements of sunlight reflected by the Earth's surface and atmosphere in the SWIR wavelengths. We use the latest release of the WFMD product (v1.8) (Schneising et al., 2023), which includes processing improvements such as an increased polynomial degree (cubic instead of quadratic) and an updated digital elevation model to account for various localized topography related biases (Hachmeister et al., 2022). Furthermore, the machine-learning based quality filter in the post-processing is improved to further reduce scenes with residual clouds. We use data with a quality flag qf = 0 (good) and

don't include data with qf = 1 (potentially bad). The WFMD product includes measurements for solar zenith angles up to 75°. We performed this analysis using data from 05.2018 to 02.2023, excluding data from the commissioning phase (11.2017 to 04.2018). Uncertainties for this data are estimated during the inversion procedure via error propagation from the spectral measurement errors given in the TROPOMI Level 1 files. Additionally, the uncertainties include a correction by statistically comparing the original uncertainties to the measured scatter relative to TCCON measurements.

## 2.2 CAMS global inversion-optimised greenhouse gas fluxes and concentrations (CAMS/INV)

The Copernicus Atmospheric Monitoring Service (CAMS) global inversion-optimised greenhouse gas fluxes and concentrations dataset (CAMS/INV) provides data for carbon dioxide, nitrous oxide and methane. The methane data is produced using the CAMS $CH_4$ Flux Inversion system (Segers et al., 2022), which is based on the TM5-4DVar inverse modeling system (Bergamaschi et al., 2010, 2013). We use release v22r1, where only ground-based observations from the NOAA network are used in the inversion (CAMS/INV-SRF) and release v21r1s, which includes satellite observations from the Greenhouse Gases Observing Satellite (GOSAT) in addition to ground-based observations (CAMS/INV-SRF-SAT). In our analysis we use the total column dry-air mole fractions and surface fluxes of methane from this dataset. The data is provided on a $2° \times 3°$ grid from 1990 to 2022. We only apply our DLM approach to the methane concentrations from this data and use the corresponding fluxes directly to help with interpretation.

## 2.3 NOAA $CH_4$ Marine Boundary Layer Reference

The Marine Boundary Layer Reference (MBLR) is a 2-dimensional matrix (time vs. latitude) created from weekly air samples from the Cooperative Air Sampling Network (Dlugokencky et al., 2021), which is created for various long-lived trace gases by NOAA–GML. The MBLR is created by first fitting the weekly data whereby the $CH_4$ level, seasonal component and short-term variations are separated. For each time step (48 evenly distributed per year) the different stations give a latitudinal distribution of $CH_4$ which is then smoothed. The global mean is calculated by averaging the smoothed latitudinal distribution for each time step. A detailed explanation can be found on the NOAA website (noa, 2022).

## 2.4 Univ. Bremen C3S/CAMS satellite data (UB–C3S–CAMS)

Annual methane increases are published by the Copernicus Climate Change Service (C3S, c3s (2023c)) in the context of the European State of the Climate (ESOTC) assessment. Here we use data from the ESOTC 2022 (c3s, 2023d) climate indicator section (c3s, 2023a). The methane data as shown on that website are (i) time series of monthly values of the column-averaged mole fraction of atmospheric methane, $XCH_4$, as derived from satellite data, and (ii) annual mean methane growth rates including uncertainty estimates as derived from this time series.

The $XCH_4$ time series corresponds to averaged satellite data over land in the latitude band $60°$ S $- 60°$ N and covers the period January 2003 to December 2022. The underlying satellite $XCH_4$ data product for 2002–2021 is XCH4_OBS4MIPS version 4.4 available from the Copernicus Climate Data Store (CDS,c3s (2023b)) website (c3s, 2018). The data product is derived from the satellite instruments SCIAMACHY/ENVISAT, TANSO-FTS/GOSAT and TANSO-FTS-2/GOSAT-2. A previous version of this data product is described in Reuter et al. (2020). This data set is extended using a year 2022 satellite-derived $XCH_4$ data product, generated for the Copernicus Atmosphere Monitoring Service (CAMS, cam (2023)) (see c3s (2023a) for details).

The combined C3S/CAMS $XCH_4$ time series has been generated by the University of Bremen (UB) and is in the following referred to as UB–C3S–CAMS data set. This data set is also used to derive annual mean methane growth rates for 2003–2022 using the method as described in Buchwitz et al. (2017), for $XCO_2$, which has later also been applied to $XCH_4$ (Reuter et al., 2020). This method provides a new time series from the monthly $XCH_4$ time series described above. The new time series is generated by computing the difference in $XCH_4$ for a given calendar month between two consecutive years (e.g., January 2019 and 2020). The time assigned to this difference is the mean time between the two months (e.g., mid-July 2019). The annual mean growth rate for a give year is the weighted average of all monthly difference values of that year.

## 3   Method

The method section is split into four parts. First we describe how the data is prepared for the DLM, meaning how we get from single observations to a time-series we can fit our model to. Next, we shortly introduce DLMs and provide information on the specific types of models we are using. In the third subsection we explain how we use an ensemble of DLMs and cross-validation to select the best model. Lastly, we describe how we estimate the bias related to imperfect satellite sampling.

### 3.1   Data preparation

The $XCH_4$ data to be used in the DLM fitting are preprocessed onto a latitude-longitude grid with sufficiently homogeneous sampling in space and time. Initially, the WFMD $XCH_4$ data product is gridded onto a $2° \times 2°$ grid. For this we assign each measurement to a single grid cell and calculate the weighted average of all measurements per cell. The measurements are weighted using the inverse measurement uncertainty to disadvantage measurements with high uncertainty. For example, reported uncertainties are higher for low albedo scenes. Thus, these scenes contribute less to the average. The coverage of the WFMD data is roughly 25% in all regions and mostly constant, except for a few days with lower coverage and the seasonal data gaps in high latitudes (see Fig. D1). To account for inhomogeneities in spatial and temporal sampling we apply the method described by Sofieva et al. (2014). This method quantifies the sampling distribution's inhomogeneity using a measure denoted as $0 \leq H \leq 1$, which is defined as a linear combination of the asymmetry $A$ and entropy $E$ of the data:

$$H = \frac{1}{2}(A + (1 - E)) \tag{1}$$

$$A = 2\frac{|\bar{x} - x_0|}{\Delta x} \tag{2}$$

$$E = \frac{-1}{log_e(N)} \sum_i \frac{n(i)}{n_0} log_e\left(\frac{n(i)}{n_0}\right) \tag{3}$$

In equation 2, the mean location of measurements is given by $\bar{x}$ (e.g. mean spatial position or mean time), $x_0$ is the central point and $\Delta x$ the width of the region. Equation 3 represents the normalized entropy with $N$ as the number of bins (e.g. grid cells or time steps), $n(i)$ the number of observations in bin $i$ and $n_0$ the sample size.

*A* can be intuitively understood as the asymmetry of the sampling distribution. For example, *A* would be high if only measurements in the eastern hemisphere are present for a given day. In contrast, *A*, would be zero if the measurements are symmetrically distributed around the central point. The normalized entropy is $E = 1$ for perfectly homogeneous sampling patterns and gets lower for each missing measurements. The entropy does however not capture the distribution of the sampling pattern. Hence, a combination of both measures is used to quantify the homogeneity of the sampling distribution. Values of $H$

close to zero indicate a homogeneous sampling distribution while values close to one indicate a very inhomogeneous distribution. Inaccurate estimates and spurious features can arise without accounting for this inhomogeneous sampling (Sofieva et al., 2014). The inhomogeneity can be calculated in the temporal domain (for each grid cell) and in the spatial domain (for each time step).

We first calculate the temporal inhomogeneity ($H_T$) for each grid cell, which quantifies how even and symmetric the data for each grid cell is distributed in the temporal domain. $H_T$ tends to be higher in cells with sparse data coverage, which are often found over the oceans and tropical rain forests due to high cloud coverage (see Fig. 1). We then filter cells with $H_T > 0.5$. This threshold value was chosen empirically to exclude cells in these regions with limited cloud-free coverage. Next, we calculate the spatial inhomogeneity ($H_S$) within the designated sub-grid, such as a zonal band. This allows to identify days with very

inhomogeneous coverage. The spatial inhomogeneity can be calculated along both spatial dimensions. Hence, we define $H_S$ as the equally weighted linear combination of the latitudinal and longitudinal spatial inhomogeneity:

$$H_S = 0.5 \cdot H_S^{lat} + 0.5 \cdot H_S^{lon} \tag{4}$$

We determine a limit, $H_S^{lim}$, as the median of $H_S$ plus two standard deviations

$$H_S^{lim} = \tilde{H}_S + 2\sigma_{H_S} \tag{5}$$

and filter out days with $H_S > H_S^{lim}$. The equation for $H_S^{lim}$ was empirically chosen and yields reliable limits for different sub-grids. Figure 1 illustrates the spatial and temporal inhomogeneity for global WFMDv1.8 data.

Finally, we compute the area-weighted average of the chosen sub-grid, generating a time series for the analysis. To further mitigate sampling bias in the global average, we first average over longitudes and subsequently over latitudes. This approach

assumes a faster mixing of background methane levels within zonal bands, while acknowledging greater latitudinal disparities. A more detailed description is given in Sec. 3.4. CAMS/INV XCH$_4$ data is already provided on a grid with complete coverage and no inhomogeneity treatment is necessary. The time series are therefore calculated using the area-weighted average of the sub-grid. Since the DLM approach is based on the assumption that errors are present and normally distributed, we add a Gaussian noise with $\sigma = 0.2 \, \mathrm{ppb}$ to the CAMS/INV XCH$_4$ time series.

## 3.2 Dynamic Linear Model fit

To extract information about the methane growth rate from the time-series we first need to calculate the underlying XCH$_4$ level, that is the smoothly changing background concentration without seasonal or short-term variation. While a simple approach,

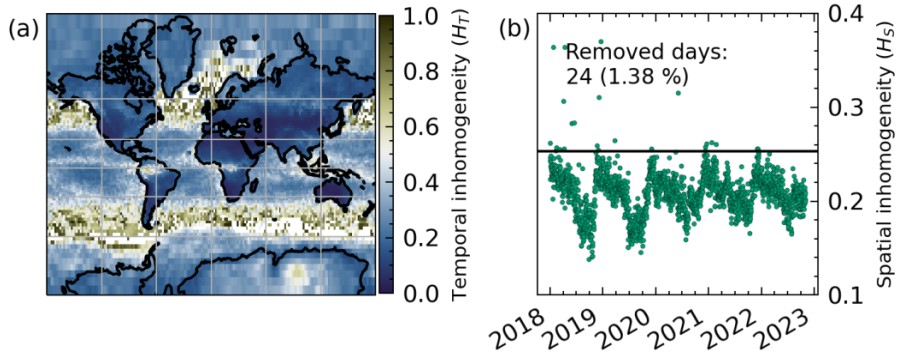

**Figure 1.** (a) Temporal inhomogeneity ($H_T$) for global $XCH_4$ WFMDv1.8 data between May 2018 and February 2023. Grid cells with $H_T > 0.5$ are omitted during analysis (b) Spatial inhomogeneity ($H_S$) for global $XCH_4$ WFMDv1.8 data. Days above the $H_S$ threshold (black line) are omitted from analysis, the threshold is set by Eq. 5 which was empirically chosen.

such as fitting a polynomial plus a trigonometric function to model the seasonality, may be considered, it is insufficient due to the complex change in $XCH_4$ levels observed in historical records (Lan et al., 2023; c3s, 2023a). The use of a moving average is not suitable due to possible data gaps, especially for high latitude bands. Therefore, we employ dynamic linear models to fit the $XCH_4$ data, which allow for the trend (i.e. the slope of the level, the growth rate) to change over time and can deal with missing data. For the analysis of global methane growth we use AMIs, similar to other relevant studies (Dlugokencky, 2022; c3s, 2023a; Schneising et al., 2023). This enables the methane growth of different studies to be readily compared. AMIs are defined as the difference in methane level between the January 1st of two consecutive years. This is a measure of the integrated growth rate over the same time span. For zonal bands we directly investigate the growth rate instead of AMIs (see Sec. 5 for a more detailed description).

A dynamic linear model is a regression model that can handle observations of varying accuracy, missing data, non-uniform sampling and non-stationary processes. It allows some of its parameters to change over time and directly models the observed variability using unobserved state variables (Laine, 2020). These DLM properties allow the analysis of not only global but also zonal methane data, which can have higher uncertainties and more gaps, especially in the higher latitudes. Additionally, the direct modeling of the data allows the partition of the signal into different components, such as an underlying level and seasonal component, which can prove advantageous beyond the scope of this paper.

A DLM can be formulated as a special case of a state-space model, that is a model which consists of some unobserved components (represented by a state vector) and the observation vector. The evolution of the state vector and the relation between observation- and state-vector are modeled by a set of equations. If these equations are linear we have a so-called dynamic linear model. The DLM we use consists of three main components. First, a slowly changing background level, which captures the long-term trend of the methane concentration. Second, a seasonal component is included to model variations arising from

seasonal cycles. This component enables variations in the phase and amplitude of the seasonal cycle to be accounted for. Third, an autoregressive component is incorporated to model noise and residual correlations in the data, accounting for short-term effects. Additionally, Gaussian noise can be included to model part of the errors. The ability of DLMs to capture changing components over time is achieved by modeling these changes as Gaussian random walks, allowing for smooth transitions and adjustments. The variances of these Gaussian random walks determine the overall variability of a certain parameter (e.g. trend). A detailed description of the model setup and the different DLM components can be found in Appendix A.

In general, the model parameters (e.g. variances, see Tab. A1 for a complete list) are not known beforehand and have to be determined. For this purpose, Maximum Likelihood Estimation (MLE) (Durbin and Koopman, 2012; Harvey, 1990) can be used. MLE is a statistical method that estimates the parameters of a model by maximizing the likelihood of the observed data given the model's assumptions. Note that the data uncertainties are not used in the MLE but are indirectly included during the data preparation (gridding) as described above. For the end-user various software packages exist which provide the implementation of this procedure, leaving only the model configuration open for the user. In our study we use the *UnobservedComponents* class of the python *statsmodel* package (Version 0.14.0, Perktold et al. (2023)), which provides the means to define a DLM and to fit it using MLE (see sta (2023) for documentation). An overview of a DLM fit for globally averaged WFMD data can be seen in Fig. 2.

DLMs have been previously used to successfully model stratospheric ozone (Laine et al., 2014), methane from different GOSAT retrievals to investigate the seasonal cycle and trend (Kivimäki et al., 2019) and methane from ground-based remote sensing (Karppinen et al., 2020). For a detailed description of DLMs, including their formulation as a special case of a state-space model, we refer readers to Durbin and Koopman (2012) and Harvey (1990). For a more concise introduction to DLMs, we refer to Laine (2020).

### 3.3 Ensemble approach and Cross Validation

The choice of model configuration is a non-trivial problem, which is impacted by prior knowledge, empirical testing and different quality measures. From prior knowledge the inclusion of a seasonal component is inferred, because the existence of a seasonality in atmospheric methane concentrations is known. Empirical testing can show that the inclusion of an autoregressive component is necessary, because the data contains residual short-term variations. The term quality measures refers to measures that facilitate model selection, such as the mean squared error (MSE), which is defined as the mean of squared differences between model and data. Additionally, the DLM provides variances (squared standard deviations) for each component which can be used to compare models; models with lower uncertainty in the level and seasonality are preferable to models with high uncertainties for these terms.

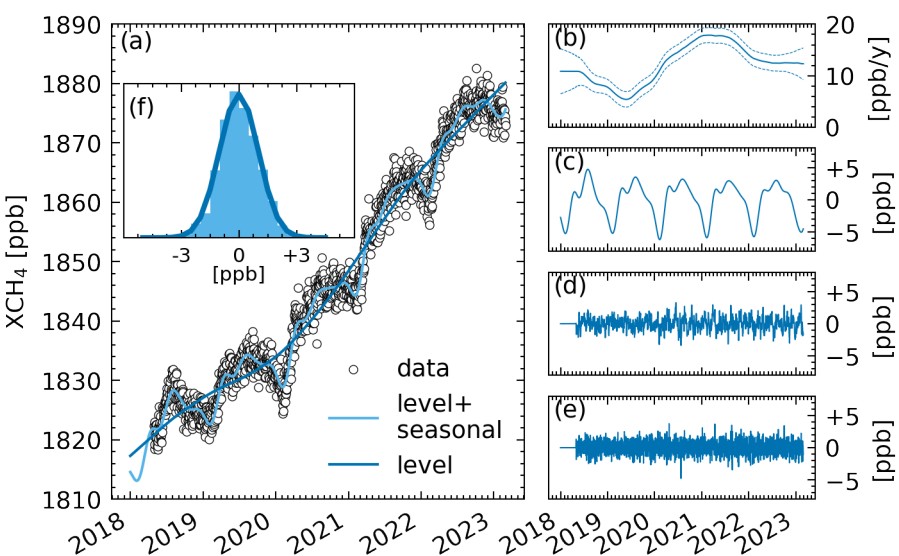

**Figure 2.** DLM fit for daily area-weighted global WFMDv1.8 data. (a) Shows the daily area-weighted global XCH$_4$ together with the level and level+seasonal components from the DLM fit (b) The trend or growth rate is the slope of the level, the dashed lines show the $1\sigma$ uncertainty, (c) Shows the seasonal component which captures the seasonal cycle, (d) The AR(1) component captures residual correlations in the data, (e) The residual shows the difference between the fit and the data, (f) A histogram of the residual shows that it is roughly normally distributed.

To avoid the need for manual model selection, we employ an ensemble approach, fitting a range of DLMs to the data and automatically selecting the best model. The ensemble consists of different DLM configurations, with varying components, as described in Appendix A. Additionally, we perform k-fold cross-validation (CV) with $k = 5$ folds for each DLM to calculate an average mean squared error (AMSE). During CV, the DLM is fitted on a portion of the data while leaving out another portion (the fold) for testing. The difference between the model fit and the fold is used to calculate the MSE, and the average MSE across all five folds per DLM provides the AMSE. Low AMSE values indicate a better model fit and help in selecting the best model for a given time-series. The final model selection is based on an aggregated score, defined as the sum of the AMSE, the variance of the level and the variance of the seasonal term:

$$S_{agg} = AMSE + \sigma^2_{level} + \sigma^2_{seas} \tag{6}$$

The inclusion of the variances ensures that the uncertainty of the level and seasonal components is considered in the selection criterion. This approach aims to select DLMs that provide good estimates of the underlying methane signal while avoiding overfitting and reliance on expert knowledge.

Different methods and measure can be used for model selection and may yield different results. We want to emphasize that the problem of model selection is non-trivial and different approaches may be suitable for different data and use-cases. Here we select the model which yields the highest certainty fit of the level and seasonal component (i.e. the XCH$_4$ signal without noise) while avoiding overfitting and manual selection. Furthermore, we want to mention that in most cases the differences

between all models in an ensemble are rather small with the best models producing approximately the same results. However, the use of a single DLM configuration for all zonal bands is not feasible due to the inherent differences in the seasonal signal for each zonal band. Additionally, an over-specified model can lead to high uncertainties in the resulting fit.

To quantify the impact of model selection, we calculate a model selection bias $\sigma^2_{model}$, which is included in the error budget

of all AMIs and growth rates. For global data we calculate the AMIs for all models in an ensemble and determine the weighted variance for each year of interest:

$$\sigma_{Model/AMI,j}{}^2 = \frac{\sum_i (AMI_{avg,j} - AMI_{i,j})^2 \tilde{\sigma}_{i,j}{}^{-2}}{\sum_i \tilde{\sigma}_{i,j}{}^{-2}}; \quad \tilde{\sigma}_{i,j}{}^2 = \sigma_{avg,j}{}^2 + \sigma_{i,j}{}^2 \tag{7}$$

where $AMI_{i,j}$ is the AMI for year $j$ and model $i$, $AMI_{avg,j}$ the average AMI of all models in year $j$, and $\sigma_{avg,j}$ and $\sigma_{i,j}$ the corresponding uncertainties as given by the DLM. For the case of growth rates we calculate a single model uncertainty which

is averaged over all time steps:

$$\sigma^2_{Model/trend} = \frac{1}{T} \sum_t \left[ \frac{\sum_i (\nu_{avg,t} - \nu_{i,t})^2 \tilde{\sigma}_{i,t}{}^{-2}}{\sum_i \tilde{\sigma}_{i,t}{}^{-2}} \right]; \quad \tilde{\sigma}_{i,t}{}^2 = \sigma_{avg,t}{}^2 + \sigma_{i,t}{}^2 \tag{8}$$

where $T$ is the total number of time steps, $\nu_{i,t}$ the growth rate for model $i$ at time step $t$, $\nu_{avg,t}$ the average growth rate of all models at time step $t$, $\sigma_{avg,t}$ the uncertainty of the average growth rate at time step $t$, and $\sigma_{i,t}$ the uncertainty of model $i$ at time step $t$ as given by the DLM.

The contribution of the model selection bias to the error budget can be seen in Tab. 1 for global AMIs and Tab. 2 for zonal growth rates. In case of global data, the contribution is small for all years with $\sigma_{Model} < 1$ ppb except for 2018, when only an incomplete time series is available. For zonal data, $\sigma_{Model}$ varies between the bands and is in the range of 1.03–4.34 ppb/y.

### 3.4   Estimation of sampling bias

The spatio-temporal coverage of S5P XCH$_4$ data is limited mostly by cloud coverage, the polar nights and poorly reflective surfaces, while additional gaps may exist due to technical problems with the satellite platform. Additionally, the sampling distribution is not completely random but is influenced by the total land mass per latitudinal band and seasonal cloud coverage over the tropical and subtropical oceans. The daily gridded data for S5P XCH$_4$ is therefore always incomplete, meaning we have some grid cells without any measurements. Here we investigate the systematic sampling bias due to polar nights, the total

effect due to sampling and the effect of the averaging method. For this, we use CAMS/INV XCH$_4$ data (see Sec. 2.2) onto which we apply different masks and averaging methods. Since CAMS/INV data has a complete coverage (due to being model

data) we can investigate the effects of different sampling masks or averaging and compare it to the results gained from the unmasked data.

We therefore compare AMIs (for global data) and growth rates (for zonal data) calculated using different sampling and/or methods. To simulate the spatio-temporal pattern of S5P sampling, we created a daily mask from gridded WFMDv1.8 data and applied it to the model data. To only simulate the systematic effect of the polar nights, we created a daily mask using the average solar zenith angle (SZA) per grid cell with a cut-off value of $75°$. The AMIs and growth rates for CAMS/INV $XCH_4$ data were calculated using the same ensemble approach used for WFMD data.

First, we investigate the effect of the averaging method. We compare standard averaging, which is defined is this study as the area-weighted mean of all grid cells in a region, with an approach we call zonal-first averaging. Zonal-first averaging, takes into account the inhomogeneous sampling at each latitude, which is influenced by the distribution of land mass and seasonal coverage. Since zonal transport occurs within weeks (Jacob, 1999), we first average the grid cells zonally, assuming that the
available data within a $2°$ band provides a good estimate of the mean $XCH_4$ at this latitude. For global data this means that we first average the data in all ninety $2°$ latitude bands and for zonal growth rates this would mean first averaging the ten $2°$ latitude bands within a $20°$ zonal band. Subsequently, we calculate the average from the zonal averages, which leads to a consistent weighting of all latitudes regardless of their individual coverage. Figure 3 shows AMIs calculated from CAMS/INV-SRF $XCH_4$ data using (a) No mask and standard averaging (b) The S5P $XCH_4$ mask and standard averaging and (c) The S5P
$XCH_4$ mask and zonal-first averaging. Using zonal-first averaging the AMIs are closer to the AMIs derived from the complete data. This is especially visible for 2020, where the AMI is overestimated by roughly 3 ppb when using the standard averaging. Therefore, we use zonal-first averaging for all calculations of globally averaged data.

Figure 4 shows zonal growth rates calculated from CAMS/INV-SRF $XCH_4$ data using (a) No mask and standard averaging
(b) The S5P $XCH_4$ mask and standard averaging and (c) The S5P $XCH_4$ mask and zonal-first averaging. Growth rates calculated on the masked data show $1\sigma$ agreement with the growth rates calculated on the complete data. Growth rates calculated using the zonal-first averaging show better agreement for the $50°–70°S$ band while for the $10°S–10°N$ band the growth rate is more variable. For all other zonal bands the results are nearly identical. Noticeably, the $70°–90°N$ band shows no variation when using the S5P $XCH_4$ mask, indicating that insufficient data coverage hinders the detection of growth rate variation. Since
sampling within a $20°$ zonal band can still vary with latitude we follow the same reasoning as in the previous paragraph and also use zonal-first averaging for zonal growth rate calculation.

The different growth rates for (a), (b) and (c) in Fig. 4 indicate the presence of a sampling bias. Thus, we investigated the effect of satellite sampling and the polar nights by applying corresponding masks to CAMS/INV $XCH_4$ data. For global data
the sampling biases are calculated by taking the squared difference between the AMI calculated on the complete data (without any sampling filtering) and the AMI calculated on the masked CAMS/INV data. We calculate a separate bias for each year in

**Table 1.** Error budget for global AMIs. $\sigma_{DLM}$ is the uncertainty provided by the DLM fit, $\sigma_{Model}$ is the uncertainty from model selection and $\sigma_{Sampling}$ is the bias due to satellite sampling. All values show $1\sigma$ uncertainties.

| Year | 2018[†] | 2019 | 2020 | 2021 | 2022 |
|---|---|---|---|---|---|
| $\sigma_{DLM}$ | 1.70 | 0.40 | 0.39 | 0.39 | 0.49 |
| $\sigma_{Model}$ | 3.22 | 0.71 | 0.32 | 0.48 | 0.35 |
| $\sigma_{Sampling}$ | 2.96 | 0.25 | 0.19 | 0.26 | 0.25 |
| $\sigma_{Sampling(SZA)}$ | 0.06 | 0.37 | 0.05 | 0.22 | 0.73 |
| $\sigma_{Total}$ | 4.70 | 0.85 | 0.53 | 0.67 | 0.65 |

All values are in ppb, [†] 2018 only includes data starting 01.05.2018.

our analysis.

$$\sigma_{Sampling}^2 = (AMI - AMI_{Sampling})^2 \tag{9}$$

$$\sigma_{SZA}^2 = (AMI - AMI_{SZA})^2 \tag{10}$$

Table 1 shows the resulting errors for global AMIs. The sampling bias is around 0.25 ppb for all fully available years and much higher for 2018 with 2.96 ppb. The contribution of the SZA related bias varies from year to year with a maximum value of 0.73 ppb in the year 2022. This variability might be related to the varying difference between the high- and mid-latitudes during the different years. When the difference between both is bigger, the masking of high-latitude regions is expected to have a larger effect on AMIs. For the analysis of zonal bands we calculate a zonal error by taking the average squared difference

between growth rates derived from the complete data and growth rates derived from the reduced CAMS/INV XCH$_4$ data for each band:

$$\sigma_{Sampling/Zonal}^2 = \frac{1}{N} \sum_i^N (\nu_i - \nu_i^{Sampling})^2 \tag{11}$$

where $\nu_i$ is the growth rate at time step $i$ and $N$ is the total number of data points. The sampling errors for zonal growth rates are shown in Tab. 2. Higher sampling errors correlate clearly with regions of high temporal inhomogeneity which further shows

the challenging sampling conditions in these regions (see Fig. 1).

## 4  Comparison of different global annual methane increases

In this section, we discuss global AMIs calculated using WFMDv1.8 data from May 2018 to February 2023. The results are shown in Fig. 5. An overall, although non-linear, rise in methane level is observed between 2019 and 2022. The most significant change occurs from 2019 to 2020, with an increase from $6.89 \pm 0.85$ ppb to $14.40 \pm 0.53$ ppb. The highest AMI is observed

in 2021, reaching $16.93 \pm 0.67$ ppb. The globally averaged XCH$_4$ has risen from $1817.32 \pm 2.81$ ppb at the beginning of 2018

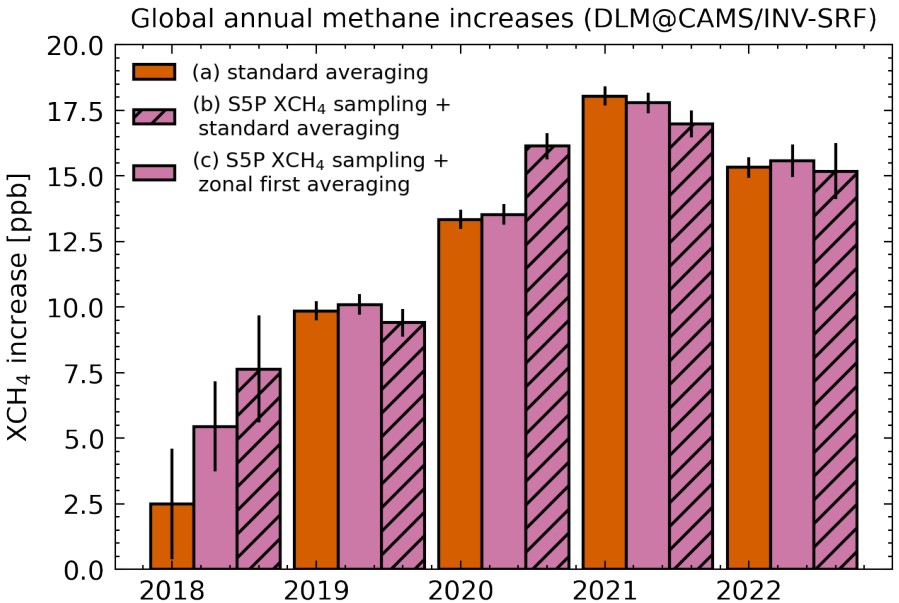

**Figure 3.** Global AMIs derived from CAMS/INV-SRF XCH$_4$ data using (a) The non-masked dataset with standard averaging, (b) The S5P sampled CAMS/INV-SRF data with standard averaging and (c) The S5P sampled CAMS/INV-SRF data using our zonal-first averaging approach.

to $1878.14 \pm 0.16 \, \mathrm{ppb}$ at the end of 2022.

To validate our findings, we compared our results with AMIs determined by Schneising et al. (2023), the NOAA–GML (Dlugokencky, 2022), and data generated for the C3S (c3s, 2023a). Additionally, we include AMIs derived using our DLM approach for monthly WFMDv1.8 data, NOAA–GML MBLR data and the UB–C3S–CAMS dataset. Table 3 and Fig. 7 provide a comparison of AMIs between 2018 and 2022. While absolute values may differ due to variations in data and methods, all AMIs exhibit the same qualitative trend. Differences are expected for various reasons. First, the difference in sampling. NOAA–GML AMIs, are based on surface flask measurements rather than satellite total column observations used in the other calculations. Second, different methods are used to derive AMIs from the data (see the corresponding sources for description the other methods). And lastly, depending on the time resolution of the data and method used, the AMIs represent either the difference between the 1st of January of two consecutive years or the difference between the monthly January average of two consecutive years. The former is the case for our DLM approach and the AMIs derived by the NOAA–GML. The latter is the case for the AMIs derived by Schneising et al. (2023) and for the C3S AMIs.

´ To discern the impact of data and methodology, AMIs for the UB–C3S–CAMS and NOAA–GML datasets created using different methods can be compared. Our DLM based AMIs of these datasets agree within $1\sigma$ with the calculations done for the C3S and by the NOAA–GML, indicating no significant differences due to the method used. However, we see a compar-

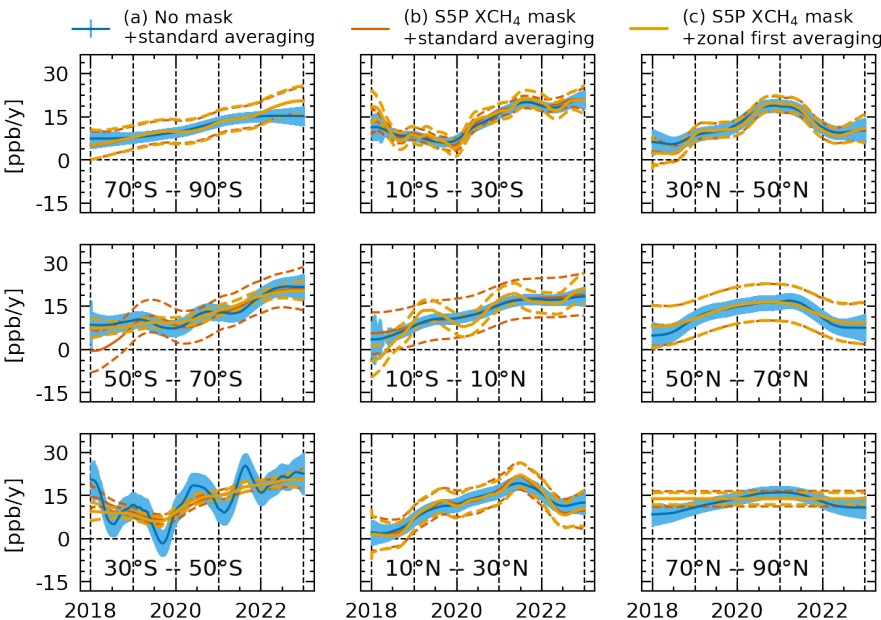

**Figure 4.** Zonal growth rates derived from CAMS/INV-SRF XCH$_4$ data using (a) The non-masked dataset with zonal-first averaging, (b) The S5P sampled CAMS/INV-SRF data with standard averaging and (c) The S5P sampled CAMS/INV-SRF data using our zonal-first averaging approach.

atively high AMI in 2022 for the combination of DLM and NOAA–GML data (see Fig. 7). This is probably related to the higher uncertainty for DLMs at the start/end of a time series, which can also be seen for UB–C3S-CAMS AMIs. A longer input time series will likely lead to a reduction in uncertainty and to a reduced deviation compared to the other 2022 AMIs. An application of our DLM approach for the complete UB–C3S–CAMS and NOAA–GML MBLR data can be seen in Appendix B.

Additionally, we used our DLM approach on monthly WFMDv1.8 data, to better compare our method to the method used by Schneising et al. (2023), which also shows agreement within $1\sigma$. Only differences smaller than $1\sigma$ are found when comparing AMIs based on the same data but different methods. We also applied our approach to CAMS/INV XCH$_4$ data (see: 2.2), for which global AMIs can be seen in Fig. 6. The AMIs are in qualitative agreement and show the same structure over the 5-year period, however significant differences in absolute values are observed for 2020 and 2022. For 2020 the CAMS/INV AMI is significantly lower compared to most other sources, while for 2022 the CAMS/INV AMI is significantly higher compared to most other sources (see Tab. 3).

The AMIs for 2020 and 2021 are the largest observed since NOAA began systematic records in 1983. The drivers contributing to these record increases have been the subject of recent debate and can be attributed to a rise in emissions, a reduction

**Table 2.** Sampling and model error for zonal growth rates. $\sigma_{DLM}$ is not included in this table since it varies with time but is shown in Fig. 8. All values show $1\sigma$ uncertainties.

| Band | $\sigma_{Model}$ | $\sigma_{Sampling}$ | $\sigma_{Sampling(SZA)}$ |
|---|---|---|---|
| 70–90° N | 1.44 | 2.70 | 0.14 |
| 50–70° N | 1.92 | 1.30 | 0.61 |
| 30–50° N | 1.87 | 1.46 | 0.00 |
| 10–30° N | 3.63 | 1.52 | 0.00 |
| -10–10° N | 3.34 | 2.65 | 0.00 |
| 10–30° S | 1.73 | 1.87 | 0.00 |
| 30–50° S | 1.03 | 3.98 | 0.00 |
| 50–70° S | 2.95 | 1.46 | 1.85 |
| 70–90° S | 4.34 | 1.94 | 0.67 |

All values are given in ppb/yr.

of the $CH_4$ sink, or a combination of both effects. According to the International Energy Agency (IEA) methane emissions from the energy sector decreased by approximately 10 % in 2020 (iea, 2021). However, additional emissions due to reduced maintenance of landfills and oil and gas infrastructure can be expected according to Laughner et al. (2021), while McNorton et al. (2022) suggest that the effect of the global slowdown on anthropogenic $CH_4$ emissions is relatively small. Some studies propose that the reduction of the OH sink, caused by decreased emissions of nitrogen oxides during the COVID-19 pandemic, may explain part of the increase (Stevenson et al., 2022; Laughner et al., 2021; Peng et al., 2022; Qu et al., 2022; Feng et al., 2023). Specifically, Stevenson et al. (2022); Peng et al. (2022) suggest that approximately half of the increase can be attributed to this effect. Conversely, several other studies attribute the majority of the $CH_4$ increase in 2020 to the growth in wetland emissions (Qu et al., 2022; Feng et al., 2023, 2022; Zhang et al., 2023).

## 5 Investigation of zonal methane growth rates

In addition to our global analysis, we investigated 20° zonal bands. The good spatio-temporal coverage of S5P $XCH_4$ might suggest that the same approach of using AMIs could be applied to identify zonal bands with anomalous methane increases. However, the impact of atmospheric transport has to be considered. While longitudinal mixing occurs on timescales of a few weeks, meridional transport is slower, taking 1-2 months between mid-latitudes and tropics or polar regions and around a year between hemispheres (Jacob, 1999; Warneck, 1999). The relatively longer atmospheric lifetime of 9.1 years (Szopa et al., 2021) compared to the mixing times, therefore guarantees a relatively even latitudinal distribution of methane in the tropo-

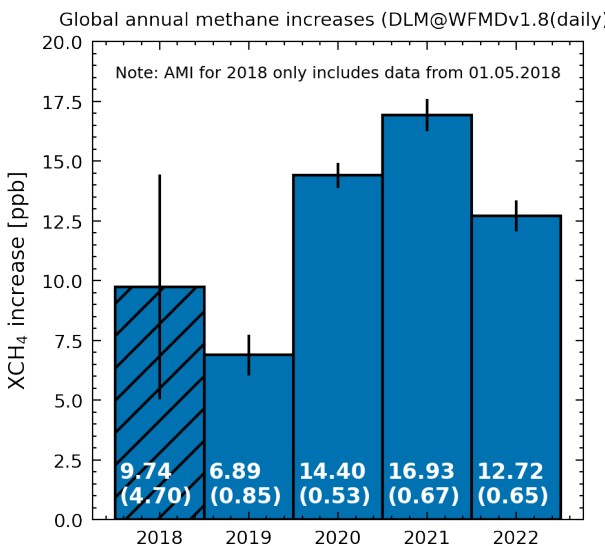

**Figure 5.** Global annual methane increases derived from Sentinel-5P/TROPOMI WFMDv1.8 data. The errorbars show the $1\sigma$ uncertainty and include the DLM, sampling and model error.

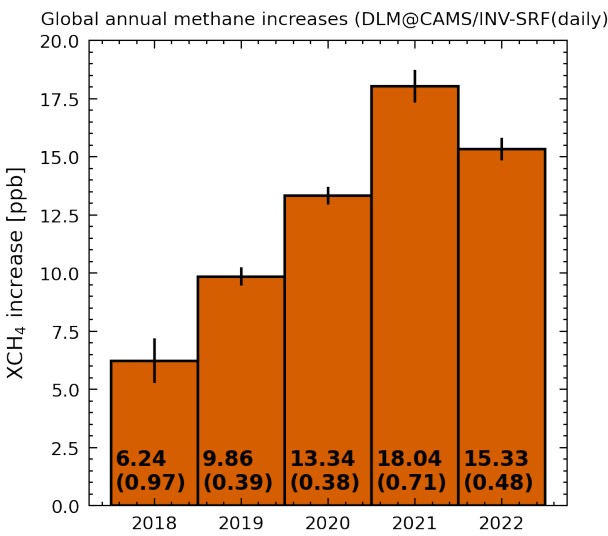

**Figure 6.** Global annual methane increases derived from CAMS global inversion-optimised greenhouse gas concentrations including only surface observations. The errorbars show the $1\sigma$ uncertainty and include the DLM and model error, which are also shown in brackets.

sphere, where the main difference is driven by the uneven distribution of $CH_4$ sources (Warneck, 1999). Thus, we need to sample at about 1 year or less to observe differences between hemispheres and 1 month or less to observe differences within a hemisphere. The daily sampling of S5P is hence faster than meridional transport, however part of the temporal information

**Table 3.** Comparison of global AMIs using different data and methods. The All errors represent $1\sigma$ uncertainties.

| Method@Dataset (time resolution) | 2018 | 2019 | 2020 | 2021 | 2022 |
|---|---|---|---|---|---|
| DLM@WFMDv1.8 (daily) | $9.74 \pm 4.70$ [†] | $6.89 \pm 0.85$ | $14.40 \pm 0.53$ | $16.93 \pm 0.67$ | $12.72 \pm 0.65$ |
| DLM@WFMDv1.8 (monthly) | $6.56 \pm 4.23$ [†] | $7.85 \pm 0.98$ | $14.39 \pm 0.93$ | $16.55 \pm 0.94$ | $12.65 \pm 1.17$ |
| Schneising et al. (2023) @WFMDv1.8 (monthly) | | $7.80 \pm 0.60$ | $15.00 \pm 1.00$ | $16.40 \pm 0.50$ | $13.90 \pm 0.60$ |
| NOAA–GML (Version 2023-10) @NOAA MBLR (daily) [*] | $8.76 \pm 0.52$ | $9.68 \pm 0.60$ | $15.16 \pm 0.41$ | $17.82 \pm 0.47$ | $13.97 \pm 0.58$ |
| DLM@NOAA MBLR (daily) [*] | $9.35 \pm 0.89$ | $8.60 \pm 0.75$ | $15.99 \pm 0.97$ | $18.16 \pm 1.22$ | $16.04 \pm 1.86$ |
| Buchwitz et al. (2017) @UB–C3S–CAMS (monthly) | $10.19 \pm 1.96$ | $9.00 \pm 2.01$ | $15.19 \pm 2.09$ | $17.09 \pm 2.09$ | $11.87 \pm 2.77$ |
| DLM@UB–C3S–CAMS (monthly) | $10.15 \pm 1.13$ | $8.92 \pm 1.30$ | $15.77 \pm 1.20$ | $17.04 \pm 1.05$ | $11.46 \pm 1.96$ |
| DLM@CAMS/INV–SURF (daily) | $6.24 \pm 0.97$ | $9.86 \pm 0.39$ | $13.34 \pm 0.38$ | $18.04 \pm 0.71$ | $15.33 \pm 0.48$ |

All values are in ppb. Uncertainties reflect one standard deviation. [†] Larger error since only data starting with May 2018 was used. [*] Input data has a weekly time resolution but AMIs are provided as for daily data (as the difference between the 1st of January of two consecutive years)

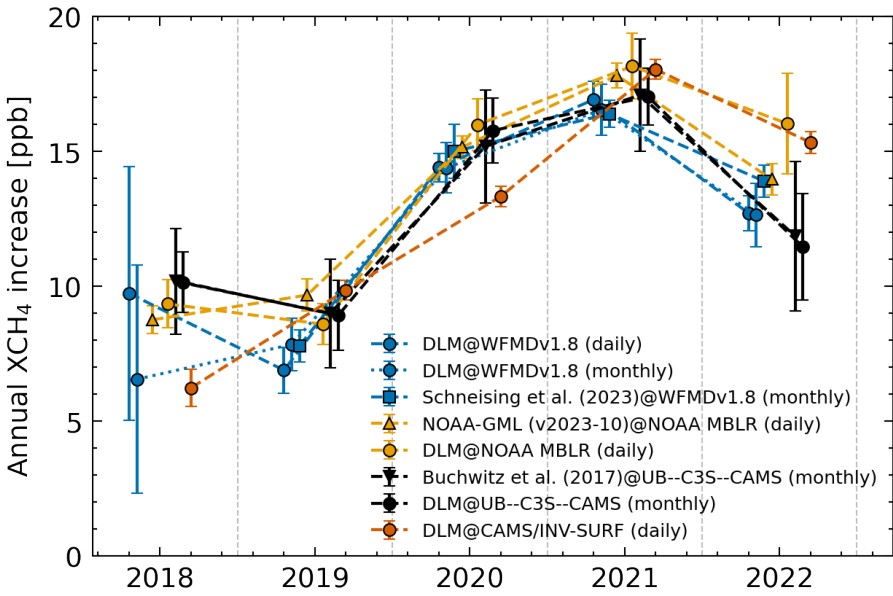

**Figure 7.** Comparison of global AMIs listed in Tab. 3. The labels are formatted as Method@Dataset. Colors indicate the type of dataset used and the markers denote the method. All errors represent $1\sigma$ uncertainties.

gets lost when using AMIs which are obtained by integrating the growth rate over one year. Thus, we investigate the growth

rate, which is the trend component of our DLM fits, to obtain a better temporal resolution of the zonal signals.

The results are shown in Fig. 8 and include growth rates derived from CAMS/INV-SRF XCH$_4$ data for comparison. The
shown errors include the uncertainty gained from the DLM fit $\sigma_{DLM}$, the model selection error $\sigma_{Model}$ and the sampling
error $\sigma_{Sampling}$ (see Sec. 3.3 & 3.4). Growth rates are similar within a hemisphere, while differences between hemispheres
are clearly visible. Additionally, no significant sub-annual variations (i.e. on a monthly timescale) in zonal growth rates are
present. Both observations are in good agreement with the known atmospheric mixing times and indicate that our data currently
allows for identification of inter-hemispheric differences while short-term variations between zonal bands are not detected. The
high latitude band between 70°–90° is included for completeness, but shows no inter-annual variability. This may be due to a
lack of real change in growth rates in this region, high uncertainties present in the data and/or the sparse data coverage. The
corresponding CAMS/INV-SRF XCH$_4$ growth rate indicates that the variability in growth rate is relatively small in this band,
which supports the first point of our explanation. We therefore exclude this band from our following discussion. Hence, we
mean bands between 10° S and 90° S when we speak of the Southern Hemisphere (SH) and bands between 10° N and 70°
N when we speak of the Northern Hemisphere (NH). The band between 10° S and 10° N represents the boundary region and
is close to the global background as can be seen in Fig. 9, which presents zonal growth rate anomalies. Zonal growth rate
anomalies are defined as the difference between the zonal and the global growth rate for each band.

Overall growth rates derived from WFMD data are close to growth rates derived from CAMS/INV-SRF XCH$_4$ data, with
an overall agreement within 1$\sigma$. The growth rates and the growth rate anomalies can be used to interpret the changes in global
AMIs and allow the identification of hemispheres or zonal bands with anomalous growth rates. Differences between the hemi-
spheres can be especially well seen in the zonal growth rate anomalies. During 2019 a decrease in growth rates can be observed
for the whole SH (except the southernmost band), while growth rates in the NH increased or stayed stable. For 2020 growth
rates for all SH bands increase strongly from roughly $0 \, \mathrm{ppb/y}$ to $20 \, \mathrm{ppb/y}$. The NH growth rates increase more slowly, except
for the 50° N – 70° N band which exhibits a small decrease in growth rate. During 2021 most zonal growth rates move towards
or around the global mean, with the strongest anomaly visible in the 10° N – 30° N band which shows some additional increase
in growth rate, which peaks mid of the year. During 2022 a clear difference between the hemispheres can be seen again, with
a decrease of growth rates in the NH and an increase or stabilization of growth rates in the SH. This difference is especially
clear when looking at the zonal growth rate anomalies in Fig. 9.

Recent studies, which discuss the record methane increases in 2020 and 2021 can help with interpreting the structure of
zonal growth rates. Peng et al. (2022) employ an atmospheric inversion using ground-based data. They attribute the increase
from 2019 to 2020 roughly equally to changes in the OH sink and an increase in wetland emissions located mainly in the NH.
In contrast, studies based on the inversion of satellite data from the Japanese Greenhouse gases Observing SATellite (GOSAT)
state that the majority of increase from 2019 to 2020 can be attributed to the African continent (Feng et al., 2023; Qu et al.,
2022) with additional increases in tropical South America in 2021 (Feng et al., 2023). Our findings can thus be seen as aligning

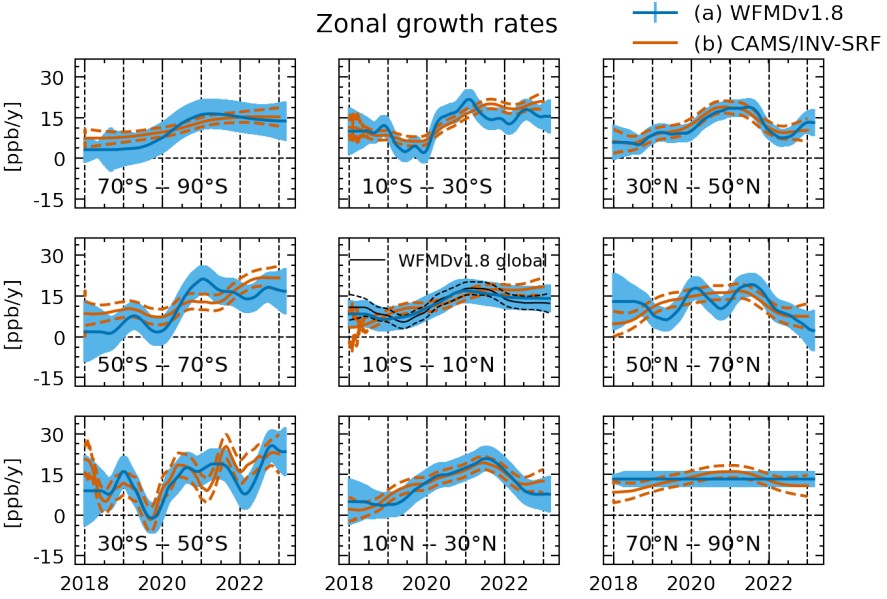

**Figure 8.** Zonal growth rates for 20° bands derived from (a) Sentinel-5P/TROPOMI WFMDv1.8 data and (b) CAMS/INV-SRF XCH$_4$ data. The errors show the 1$\sigma$ uncertainty.

with recent studies by Feng et al. (2023); Qu et al. (2022). The increase in SH growth rates from 2019 to 2020 is consistent with increased wetland emissions. The rise in the NH latitudinal bands during 2020 can be explained by the decreasing OH sink, primarily located in the NH (Peng et al., 2022; Feng et al., 2023). However, the continued increase in 2021 cannot be

solely explained by the OH sink, as OH levels mostly recovered in that year according to Feng et al. (2023) and (Peng et al., 2022). Possible explanations for the ongoing increase are persistent wetland emissions Feng et al. (2023), as well as the return to pre-pandemic methane emissions form the energy sector in 2021 (iea, 2023). Finally, the decrease of growth rates in the NH and increase of growth rates in the SH during 2022 hasn't been discussed to our knowledge. Our results therefore indicate that the decrease in global AMI from 2021 to 2022 can be attributed to a reduced growth rate in the NH. We further investigate this

in the next section.

## 6 CAMS/INV Fluxes

As mentioned before, zonal growth rates provide information about the change of methane concentration in a given zonal band, including changes in sources, sinks and transport patterns. These transport patterns would average out for global AMIs given a perfect coverage. In Sec. 3.4 we applied a S5P XCH$_4$ mask to CAMS/INV XCH$_4$ data (see Fig. 3) and compared AMIs

calculated from this masked data with AMIs calculated from the complete data. Since differences between these AMIs are small, we conclude that the effect of related sampling biases is limited. Therefore, changes in global AMIs can be attributed

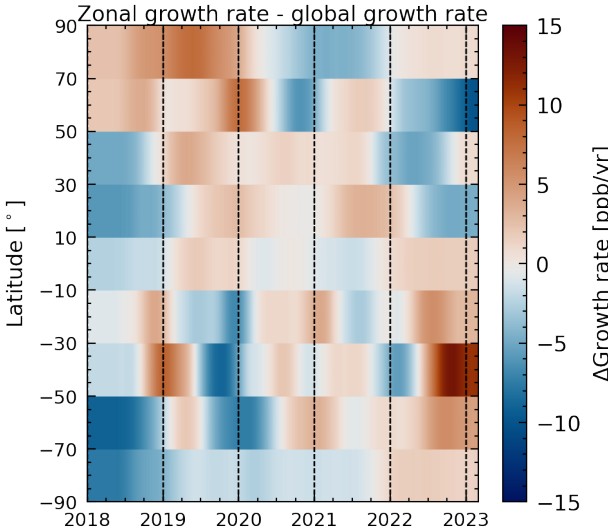

**Figure 9.** Zonal growth rate anomalies for $20°$ bands derived from Sentinel-5P/TROPOMI WFMDv1.8 data. The anomalies are defined as the difference between zonal and global growth rates.

to the total source-sink balance of methane and not to changes in transport patterns. Whether this is also true for zonal growth rates is less clear, since transport effects are expected to be stronger especially within hemispheres.

The agreement within $1\sigma$ of zonal growth rates derived from WFMD and CAMS/INV $XCH_4$ data in Fig. 8, suggest that the structures observed in our zonal growth rates are not artifacts from sampling related biases. However, we cannot rule out transport effects from this comparison, meaning we can't clearly attribute changes in hemispheres or zonal bands over the years to a change in the source-sink balance. Hence, we also investigated the change of surface fluxes between consecutive years which are readily available for the CAMS/INV data. In Figure 10, 11 and 12 we present total, wetland and non-wetland
fluxes from CAMS/INV-SRF data respectively. The category of non-wetland fluxes includes all other anthropogenic emissions as well as contributions from oceans, wild animals, the soil sink, termites and biomass burning.

Large changes in fluxes are identified between all the years investigated. The wetland flux difference between 2019 and 2020 indicate a strong increase in the NH as indicated by Peng et al. (2022) as well as some increase in the SH wetlands as reported by
450 Feng et al. (2023) and Qu et al. (2022). We expect the SH wetland fluxes to be underestimated as indicated by Feng et al. (2023) because the CAMS/INV-SRF data is based on ground-based measurements from the NOAA network, similar to the inversion performed by Peng et al. (2022), due to the poor coverage in the tropics. Interestingly, wetland fluxes from CAMS/INV-SRF-SAT data including satellite measurements from GOSAT show stronger SH wetland emissions between 2019 and 2020 as shown in Figure 13. Additionally, an increase in non-wetland fluxes occurs between 2019–2020 and 2020–2021. In the first
case these increases are mainly focused on China, while in the second case additional increases over the Indian subcontinent

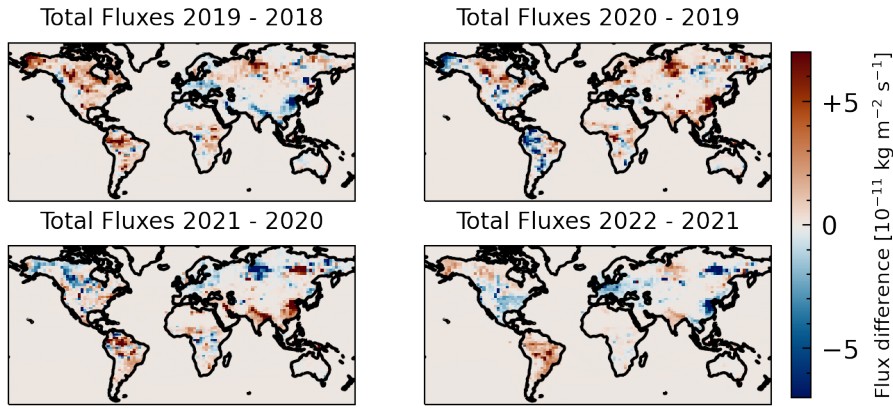

**Figure 10.** Difference between total surface fluxes from the CAMS/INV-SRF data.

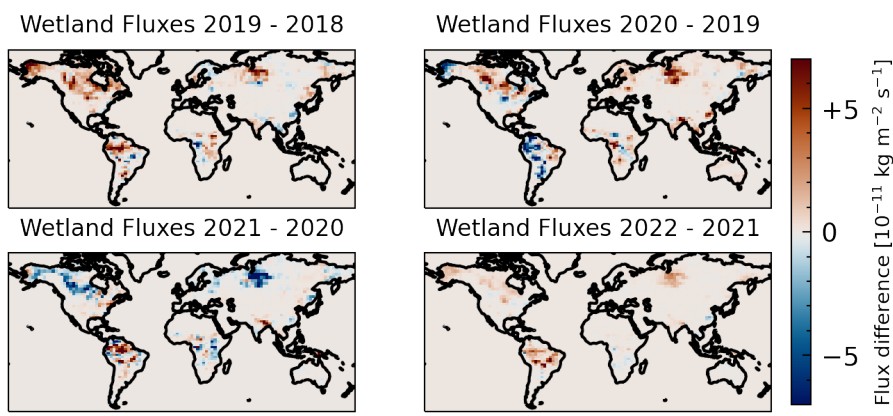

**Figure 11.** Difference between wetland surface fluxes from the CAMS/INV-SRF data.

can be seen. Between 2021 and 2022 a clear decrease of total surface fluxes can be seen in large parts of the NH, while strong increases can be observed over the whole of South America. The large decreases in the NH can be clearly attributed to changes in the non-wetland fluxes, while the increase over South America seems to involve a combination of wetland and other fluxes. Therefore, CAMS/INV fluxes imply that the changes in zonal growth rates we observed both in WFMD and CAMS/INV are not merely due to changes in transport patterns but correlate with changes in surface methane fluxes between the years. This conclusion is strengthened by the qualitative agreement of the flux changes between CAMS/INV-SRF and the aforementioned studies by Peng et al. (2022); Feng et al. (2023); Qu et al. (2022). However, further research is needed to substantiate these inferences.

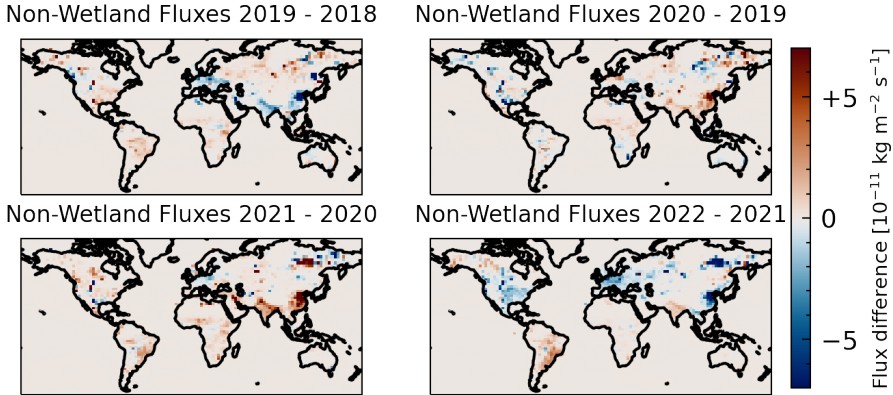

**Figure 12.** Difference between other surface fluxes from the CAMS/INV-SRF data.

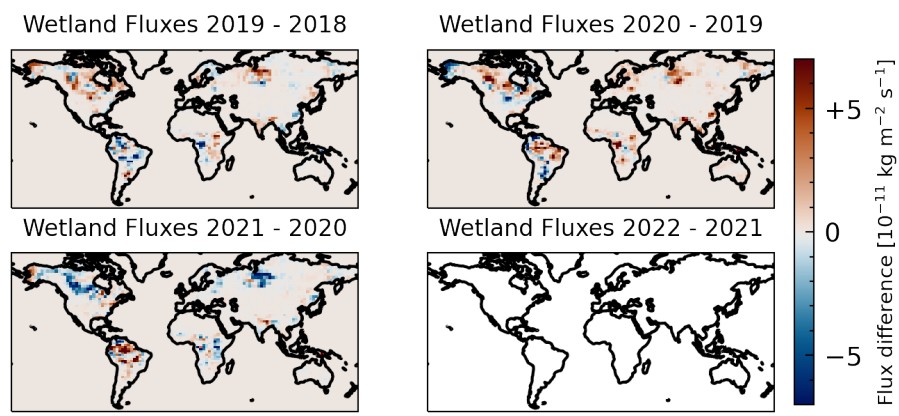

**Figure 13.** Difference between total surface fluxes from the CAMS/INV-SRF-SAT data. No Flux difference is available for 2022 - 2021, since the dataset currently ends in 2021.

## 7 Conclusions

In this study, we presented a DLM based approach to calculate methane growth rates and AMIs from S5P/TROPOMI data. We addressed sampling-related biases by comparing AMIs and growth rates derived from CAMS/INV $XCH_4$ data both with and without S5P $XCH_4$ sampling. Further, we included a bias related to the model selection in our error budget. Our calculations of global AMIs based on WFMDv1.8 data from 2018 to 2022 demonstrate good agreement with other AMIs. Additionally, we separated the influence of the fitting method and the underlying data by applying our DLM approach to other datasets. We show that using the same input data results in agreement within $1\sigma$ between all AMIs. Using the same method but different input data results in qualitative agreement but with differences larger than $1\sigma$. Nevertheless, the consistency of AMIs derived from diverse datasets, such as ground-based data from NOAA and dry-air mole fraction data from WFMDv1.8 and UB–C3S–

CAMS, highlights the robustness of these various approaches. The record methane increase in 2020 and 2021 is therefore well identified in the different data sets, which use different methods to assess AMIs. The underlying factors driving these increases, as discussed in Section 4, remain however a subject of debate.

In addition to global AMIs we investigated growth rates for $20°$ zonal bands which provide spatial information to the global AMIs. We argue that this is possible due to (a) the faster zonal mixing in comparison to meridional mixing and (b) the faster satellite sampling in comparison to the meridional mixing times. Firstly, comparisons of zonal growth rates from S5P/TROPOMI data with growth rates from CAMS/INV-SRF $XCH_4$ data show agreement within $1\sigma$. Additionally, we investigated growth rates calculated from CAMS/INV-SRF $XCH_4$ data filtered using a S5P $XCH_4$ mask which indicate that no significant sampling biases exist for the zonal band approach. Still we want to emphasize that meridional transport can affect the zonal growth rates, meaning they don't necessarily indicate changes in the sources and sinks of methane but might also show systematic changes in transport patterns. The zonal growth rates exhibit clear differences between the hemispheres for 2019 and 2022, whereas growth rates are more similar for 2020 and 2021. Differences within a hemisphere are mostly smaller and no additional short-term variations are visible, which might reflect the well-mixed state of the atmosphere within a hemisphere. The low growth rates in the SH in 2019 and subsequent increases suggest a rise in atmospheric methane in that region, possibly driven by tropical wetland emissions according to Feng et al. (2023); Qu et al. (2022); Zhang et al. (2023). Other factors potentially contributing to these changes include variations in the OH sink due to pollutant reductions during the COVID-19 pandemic (Feng et al., 2023; Qu et al., 2022; Peng et al., 2022) and the changes in global methane emissions due to the COVID-19 pandemic and the subsequent recovery (iea, 2021).

We further investigated this inter-hemispheric differences by investigating the surface fluxes available from CAMS/INV data. We argue that this is possible since (a) growth rates derived from WFMDv1.8 data are similar to growth rates from CAMS/INV data and (b) no significant sampling bias is present as we showed in Sec. 3.4. The total surface fluxes show clear changes between the years and the partition into wetland and non-wetland (mainly anthropogenic) fluxes allows further interpretation. Furthermore, changes in fluxes show reasonable qualitative agreement with findings reported by Feng et al. (2023); Qu et al. (2022); Peng et al. (2022).

In addition to the confirmation of known results, new conclusions are also drawn. Most notably, the decrease of the global AMI in 2022 is caused by reduced NH zonal growth rates. This is clearly visible in zonal growth rates derived from S5P $XCH_4$ and CAMS/INV $XCH_4$ data. Investigation of the corresponding model surface fluxes, indicates that changes in zonal growth rates are consistent with the decrease of non-wetland fluxes in the NH and the continuing increase of wetland fluxes in the SH. However, more research is needed to substantiate this inferred connection between the change in NH growth rates and fluxes.

In summary, our DLM-based approach allows calculating growth rates or AMIs for global and zonal S5P/TROPOMI data. This approach is computationally inexpensive and readily allows for the constant integration of new data, enabling timely as-

sessments of global methane concentration changes. Importantly, no additional prior information about the atmospheric state is required. We believe that our approach provides an additional valuable tool for investigating atmospheric methane concentrations, enabling the rapid identification of regions of interest, such as the 2022 NH. Furthermore, our approach can be readily applied to other datasets facing similar challenges, such as inhomogeneous sampling, non-linear trends, and data gaps. For the 70°–90° N band our method failed to identify any changes in growth rate, however this result is in good agreement with the growth rates from CAMS/INV-SRF XCH$_4$ data which themselves only show small variations. This indicates that (a) the small changes in growth rate could not be distinguished from the random variability in the data or (b) no anomalous increases in growth rate are visible for the northern high latitude regions in the observed period from 2018 – 2022.

Future research could aim to improve this approach, especially for high latitude regions, to identify smaller changes in growth rates. Better estimates of the impact of meridional transport on zonal growth rates could help to provide better error estimates for our method. The 2022 decrease in NH growth rates could be investigated in more detail and this approach be extended to include data sets of other atmospheric constituents. Data from future satellite missions, with lower uncertainties and increased data coverage, could enable the investigation of sub-annual changes in growth rates, which are presently not detectable. Finally, zonal growth rates of long-lived gases (e.g. HF) without any significant sources or sinks could possibly enable the quantification of atmospheric transport patterns.

*Code and data availability.* CAMS global inversion-optimized greenhouse gas fluxes and concentrations are available from https://ads.atmosphere.copernicus.eu/. Sentinel-5P/TROPOMI WFMD data is available from https://www.iup.uni-bremen.de/carbon_ghg/products/tropomi_wfmd/. NOAA MBL data is available from https://gml.noaa.gov/ccgg/mbl/. Example code to recreate Fig. 5 (Global annual methane increases), including the gridding and processing of the data, is available under https://doi.org/10.5281/zenodo.8178927.

## Appendix A:  Model setup and ensemble size

The structure of our DLMs assumes that the measured methane signal can be separated into a slowly changing background level, a seasonal component and noise term. This section closely follows the more detailed description in Durbin and Koopman (2012) and Harvey (1990).

The level component can be described by the following formulas

$$\mu_{t+1} = \mu_t + \nu_t + \epsilon_{t,\,level}, \qquad\qquad \epsilon_{t,\,level} \sim N(0, \sigma^2_{level}) \qquad\qquad\qquad (A1)$$

$$\nu_{t+1} = \nu_t + \epsilon_{t,\,trend}, \qquad\qquad \epsilon_{t,\,trend} \sim N(0, \sigma^2_{trend}) \qquad\qquad\qquad (A2)$$

where $\mu_t$ is the level, $\nu_t$ is the trend (i.e. the slope or growth rate) and $\epsilon_t$ are steps in a random walk sampled from a Gaussian distribution. The random walks allow components to change over time. Since we want to allow for a smoothly changing level

we allow the trend to change over time. Additionally, we enforce a constraint of zero variance for the level to ensure that short-term fluctuations in the background level are not allowed:

$$\sigma^2_{level} = 0 \tag{A3}$$

$$\sigma^2_{trend} > 0 \tag{A4}$$

The seasonal part of the signal is modeled by a truncated Fourier-series with $h$ harmonics:

$$\gamma_t = \sum_{j=1}^{h} \gamma_{jt} \tag{A5}$$

with

$$\gamma_{j,t+1} = \gamma_{jt} cos(\lambda_j) + \gamma^*_{jt} sin(\lambda_j) + \epsilon_{seas}, \quad \epsilon_{seas} \sim N(0, \sigma^2_{seas}), \quad \lambda_j = \frac{2\pi j}{s} \tag{A6}$$

$$\gamma^*_{j,t+1} = -\gamma_{jt} sin(\lambda_j) + \gamma^*_{jt} cos(\lambda_j) + \epsilon^*_{seas}, \quad \epsilon^*_{seas} \sim N(0, \sigma^2_{seas}), \quad j = 1, ..., h \tag{A7}$$

where $s$ describes the seasonality of the data, e.g. for monthly data $s = 12$ or for daily data $s = 365$ when modeling yearly patterns. The value of $s = 365.25$ can be used to account for leap years. We use $s = 365.2$ which is equal to the average number of days per year between 2018 and 2022. For $\sigma^2_{seas} > 0$ the seasonal cycle is allowed to change over time. We allow values of $h \in \{1, 2, 3, 4\}$, to account for varying levels of complexity in the seasonal cycle. The motivation for this is twofold. Firstly, we want to only model the basic structure of the seasonal cycle and not the whole signal. Secondly, the inclusion of more harmonics introduces further parameters which have to be estimated. This quickly leads to high uncertainties in the produced fit since not enough information is included in the data to account for the growing number of parameters.

The noise term accounts for residual correlations as well as random Gaussian noise in the signal. Residual correlations can be modeled by an autoregressive component which includes a serial dependence between the observations. An autoregressive noise of order $n$ includes a memory of the last $n$ measurements. For $n = 1$ this $AR(1)$ term is

$$\eta_{t+1} = \rho \eta_t + \epsilon_{AR(1)}, \quad \epsilon_{AR(1)} \sim N(0, \sigma^2_{AR(1)}), \rho \in [0, 1] \tag{A8}$$

where $\eta_t$ is the autoregressive component, $\rho$ determines the strength of the autocorrelation (the memory of the previous time step) and $\epsilon_{AR(1)}$ is again a step in a Gaussian random walk. This component introduces the parameters $\rho$ and $\sigma^2_{AR(1)}$ to the model. We confine our autoregressive component to the order of $n = 1$, which is enough to model the residual correlations in our data. Higher orders would introduce further parameters to be estimated and lead to a harder interpretability of the results. However, exclusion of the AR(1) component leads to bad fits since the model fails to account for the data variability.

An additional Gaussian noise can be included

$$\epsilon_{irr}, \quad \epsilon_{irr} \sim N(0, \sigma^2_{irr}) \tag{A9}$$

**Table A1.** DLM parameters.

| Parameter | Description | Allowed range |
|---|---|---|
| $\sigma^2_{level}$ | Variability of level | 0 |
| $\sigma^2_{trend}$ | Variability of trend | $[0, \infty]^*$ |
| $\sigma^2_{seas}$ | Variability of seasonal cycle | 0 or$^\dagger$ $[0, \infty]^*$ |
| $\rho$ | Parameter of AR(1) | $[0, 1]^*$ |
| $\sigma^2_{AR(1)}$ | Variability of AR(1) | $[0, \infty]^*$ |
| $\sigma^2_{irr}$ | Variability of Gaussian error | 0 or$^\dagger$ $[0, \infty]^*$ |
| $h$ | Number of harmonics | $1 - 4^\dagger$ |

$^*$ determined by maximum likelihood estimation during DLM fit, $^\dagger$ Different
settings are part of ensemble

which we call *irregular* component.

The complete signal can be then written as the sum of these components

$$y_t = \mu_t + \gamma_t + \eta_t + \epsilon_{irr} \tag{A10}$$

The ensemble size is determined by the number of possible model configurations. In our case this is determined by whether to allow variability of the seasonal cycle, whether to include a Gaussian error term and the number of harmonics: $N = 2 \cdot 2 \cdot 4 = 16$. An overview of all parameters can be found in Tab. A1.

## Appendix B: Replication of complete NOAA GML & UB–C3S AMIs

To investigate the effect of the fitting method on AMIs we replicated AMIs calculated by the NOAA–GML and C3S in Fig. B1 and B2 respectively. Here we present the comparison for the whole available time range (subset of data can be seen in Tab. 3).

## Appendix C: Global AMIs and zonal growth rates derived from CAMS global inversion-optimised greenhouse concentrations

Here we present global AMIs and zonal growth rates for CAMS/INV-SRF-SAT data which includes satellite measurements from GOSAT in its optimization (see Fig. C1 and C2).

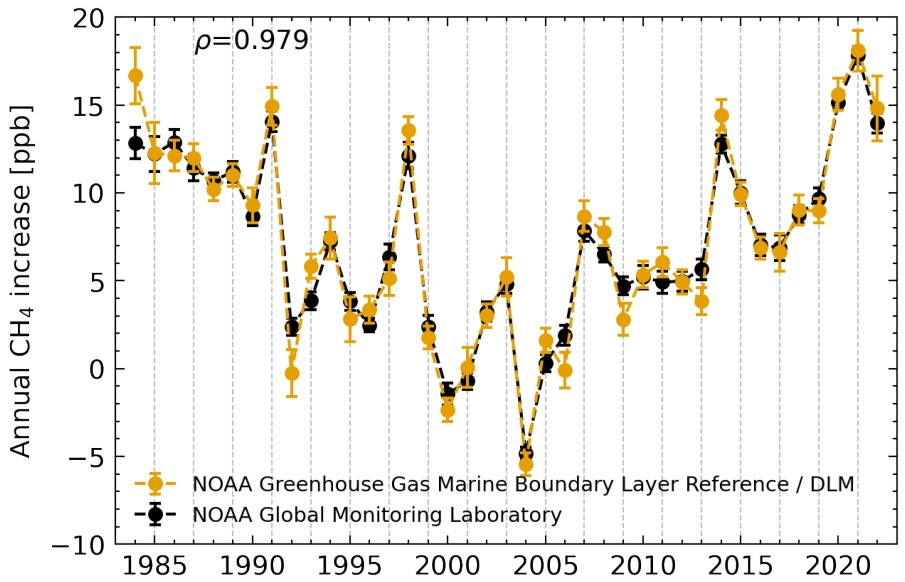

**Figure B1.** Comparison of global annual methane increases derived from the NOAA–GML MBLR data using different methods.

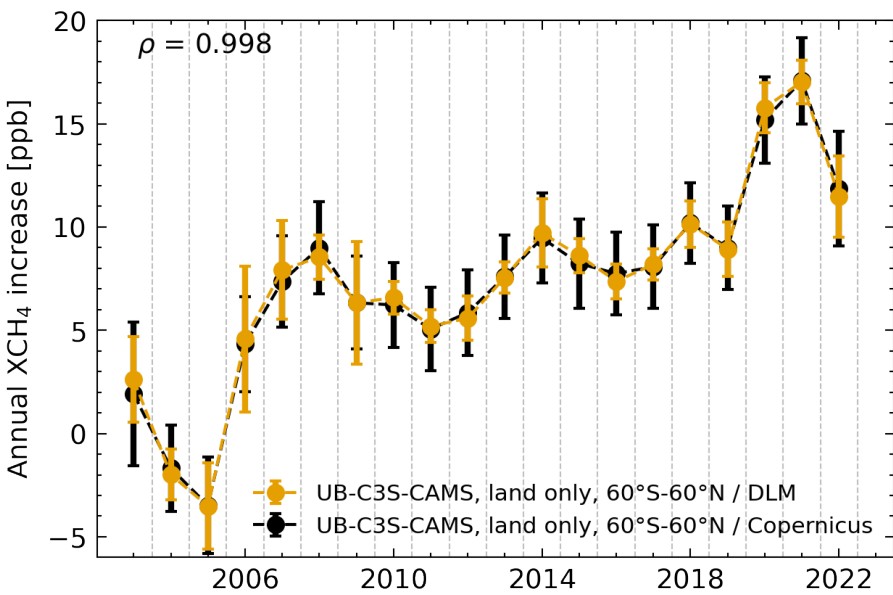

**Figure B2.** Comparison of global annual methane increases derived from C3S XCH4_OBS4MIPS v4.4 data which is extended by CAMS NRT data after 2021 using different methods.

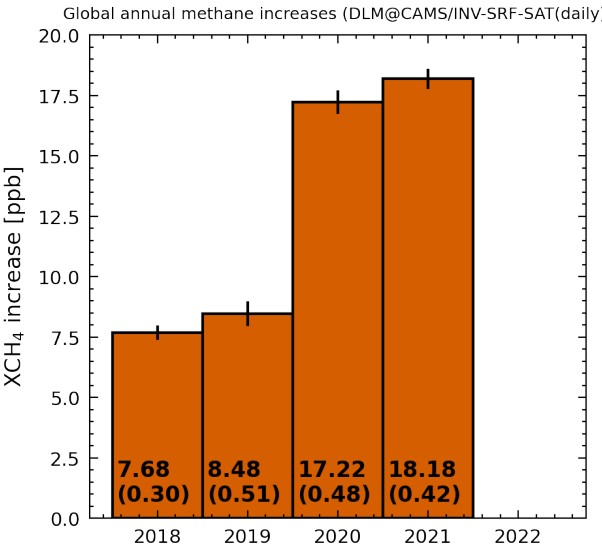

**Figure C1.** Global annual methane increases derived from CAMS global inversion-optimised greenhouse concentrations including both surface and satellite observations.

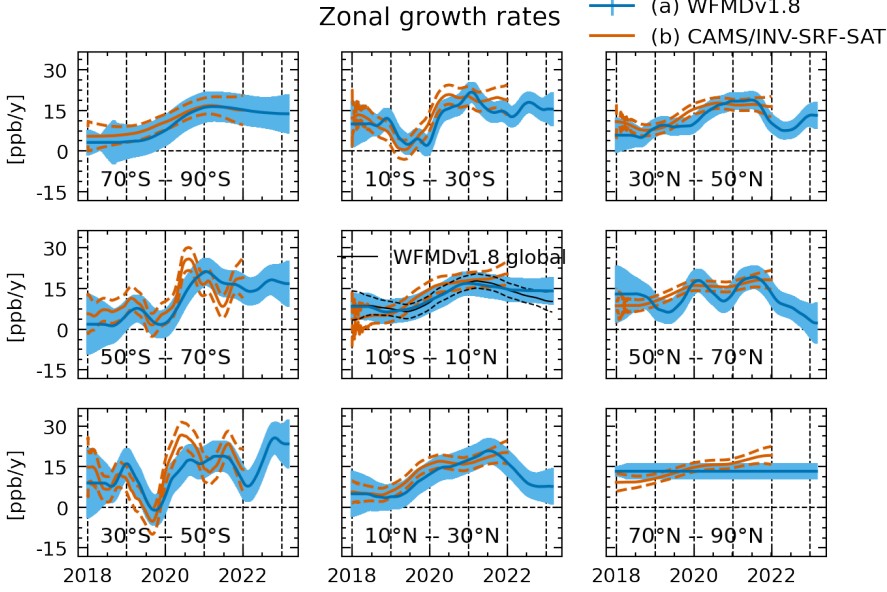

**Figure C2.** Zonal growth rates for $20°$ bands derived from (a) Sentinel-5P/TROPOMI WFMDv1.8 data and (b) CAMS/INV-SRF-SAT data. The errors show the $1\sigma$ uncertainty.

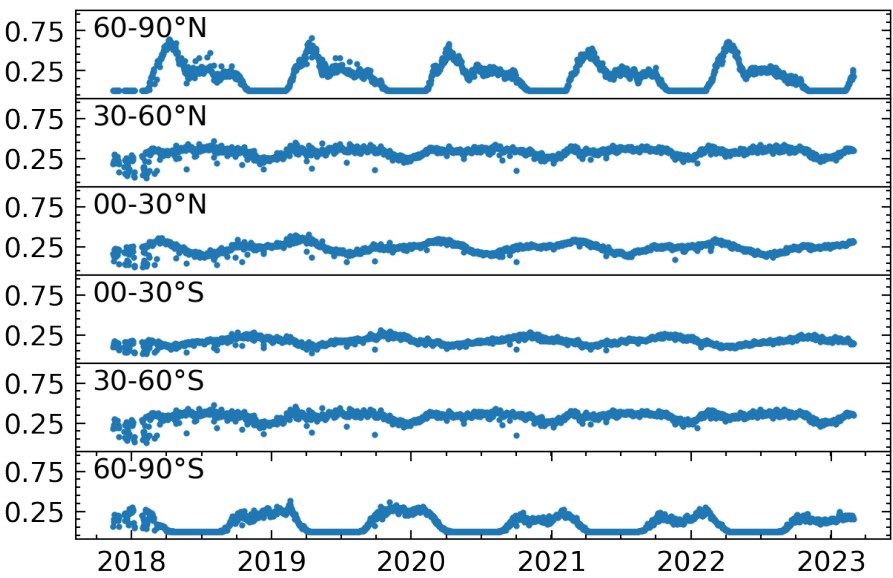

**Figure D1.** Area-normalized coverage of S5P/TROPOMI WFMDv1.8 data for 30° zonal bands, representing the high latitude, mid-latitude and tropic regions. For the calculation data on a 2°×2° grid was used. The coverage is given on a scale from 0 to 1, where 1 represents complete coverage within a band.

## Appendix D: WFMDv1.8 data coverage

Figure D1 shows the area-normalized coverage of S5P/TROPOMI WFMDv1.8 data. It can be seen that tropics, mid-latitudes
and high latitudes have mostly an average coverage of about 25%, while the high latitude regions have seasonal gaps due to the polar night.

*Author contributions.* JH developed the methodology and conducted the formal analysis. OS provided information on the WFMDv1.8 data. MiB provided information on the UB-C3S-CAMS data. JPB provided valuable input on atmospheric transport effects. MaB provided supervision and helped with the conceptualization. JH wrote the initial draft of this manuscript. All authors contributed significantly to the
conception of the analysis, jointly discussed the results and provided constructive comments to improve the manuscript.

*Competing interests.* The authors declare that they have no conflict of interest.

*Disclaimer.*

*Acknowledgements.* This publication uses and contains modified Copernicus Atmosphere Monitoring Service information (2018-2022) and modified Copernicus Sentinel data (2018–2023). Sentinel-5 Precursor is an ESA mission implemented on behalf of the European Commission. The TROPOMI payload is a joint development by the ESA and the Netherlands Space Office (NSO). The Sentinel-5 Precursor ground-segment development has been funded by the ESA and with national contributions from the Netherlands, Germany, and Belgium. The pre-operational TROPOMI data processing was carried out on the Dutch national e-infrastructure with the support of SURF Cooperative. Scientific color maps (Crameri, 2021) are used in this study to prevent visual distortion of data and exclusion of readers with color-vision deficiencies (Crameri et al., 2020).

*Financial support.* This project is funded by the State and University of Bremen. In particular the University of Bremen funding for the junior research group 'Greenhouse gases in the Arctic' is acknowledged by JH and MB. We gratefully acknowledge the funding by the Deutsche Forschungsgemeinschaft (DFG, German Research Foundation) – Projektnummer 268020496 – TRR 172, within the Transregional Collaborative Research Center "ArctiC Amplification: Climate Relevant Atmospheric and SurfaCe Processes, and Feedback Mechanisms (AC)[3]". We also acknowledge the received funding from the European Space Agency (ESA) via the projects GHG-CCI+ and MethaneCAMP (ESA contract nos. 4000126450/19/I-NB and 4000137895/22/I-AG) and from the german ministry of education and research (BMBF) within its project ITMS via grant 01 LK2103A. The TROPOMI/WFMD version 1.8 data have been generated using funding from ESA (GHG-CCI and MethaneCAMP projects). The TROPOMI/WFMD retrievals presented here were performed on HPC facilities of the IUP, University of Bremen, funded under DFG/FUGG grant nos. INST 144/379-1 and INST 144/493-1. Part of the AMIs presented were generated for Copernicus ESOTC 2022 (https://climate.copernicus.eu/climate-indicators/greenhouse-gas-concentrations) funded by EU Copernicus Climate Change Service via project C3S2_ 312a_ Lot2.

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
