# Peer review of "Zonal variability of methane trends derived from satellite data"

_EGUsphere, 2023_

## Author Comment (AC1)

**Author reply**

Jonas Hachmeister

07.11.2023

**1  Answer to the editor**

We would like to thank the editor and the reviewers for their careful reading and reviewing of our manuscript and for their helpful comments and constructive criticism, which have helped us to improve the quality of the manuscript.

We performed some major revisions on the manuscript to make it more comprehensible and easier to read. To this end we rephrased some sections and divided each section into multiple paragraphs. In addition to the changes motivated by the comments we made the following changes:

- We updated Section 3.4 by adding further investigation of sampling bias for zonal growth rates, this includes a new figure, showing growth rates for CAMS/INV data using different averaging methods and with/without S5P $XCH_4$ mask (Fig 4 in the new manuscript).

- We extended our calculations of zonal growth rates by using the alternative of zonal first averaging (as we use for the global analysis). This is motivated by the changes to section 3.4 and to provide consistency between our analysis of global and zonal data. Note that (i) There are no qualitative changes to our results and hence no changes to our conclusions (ii) This leads to the quantitative change of $\sigma_{model}$ in Section 3.3. (see Tab. 2) and an updated Figure 4,8 and 9.

- Additionally, we changed the Gaussian noise added to the CAMS/INV data from $\sigma = 0.5$ to $\sigma = 0.2$ ppb. Together with the above-mentioned change to zonal first averaging for zonal data, this leads to some quantitative changes in growth rates of CAMS/INV data. As explained above we note that these changes do not lead to qualitative changes in the results and hence to no changes in our conclusions.

- In Section 4 we changed our description of the differences between CAMS/INV AMI and the other AMI because it contained a misunderstanding with respect to data from some years.

- We updated the NOAA-GML AMIs by using the latest available data. This enables us to provide a 2022 AMI for DLM@NOAA MBLR data (see Tab. 3 and Fig. 7). Note that this leads to no significant changes of our results and hence to no changes in our conclusions.

In the following we list the reviewer comments in **bold** font and our answers in normal font.

**2 Answer to Reviewer 1**

**2.1 General comments**

**Referee comment: Description of data: I was confused by the strengths and limitations of the TROPOMI data used. Given the time spent on quantifying the errors in the growth rates due to incomplete coverage, this seems significant.**

1. **Referee comment: I was curious about the distribution of the data. For example, how much area-normalized TROPOMI data is available in the high latitudes and tropics compared to the mid latitudes? This would help me understand the usefulness of the inhomogeneity metrics.**

   Author's reply: We added a figure to the appendix (Fig. D1) which shows the area-normalized distribution of data points for the high latitudes, mid-latitudes and tropics. The coverage is around 25% for all regions, except for the seasonal gaps in the polar region. Due to this relatively low data coverage the homogeneity metrics are useful to filter out (a) days with inhomogeneous spatial coverage (b) regions with inhomogeneous temporal coverage (see Fig. 1).

2. **Referee comment: I was also curious about the potential role of systematic biases in the TROPOMI data. The authors use the WFMDv1.8 product described by Schneising et al. (2023), but to my knowledge, this product has not been specifically validated for snow and ice covered scenes (an ongoing source of bias in other retrievals).**

   Author's reply: Great efforts were made to minimize systematic biases in the TROPOMI/WFMD product. The improvements incorporated into the current version v1.8 include better consideration of possible surface spectral albedo variations within the fitting window, updating the digital elevation model to minimize topography-related biases, refinement of the machine-learning-based quality filter and bias correction, and efficient orbit-wise destriping based on combined wavelet–Fourier filtering. The validation presented in Schneising et al. (2023) includes all available TCCON sites, also those with variable seasonal snow cover: There are no indications of a systematic snow or ice bias in the WFMD product. An independent analysis for the years 2018-2021 within the MethaneCAMP project has also shown that there is no systematic seasonal dependence of the difference of TROPOMI/WFMD to TCCON at the 4 Arctic stations (only partially in 2019 at three of them). This means that a common bias, dependent on snow and ice cover is improbable.

3. **Referee comment: Finally, I was curious how the authors define uncertainty for each of these datasets.**

   Author's reply: In the case of TROPOMI/WFMD, the original uncertainties are estimated during the inversion procedure via error propagation from the spectral measurement errors given in the TROPOMI Level 1 files. These estimates underestimate the actual uncertainties as the (unknown) pseudo-noise component determined by specific atmospheric parameters or instrumental features is not considered. The reported uncertainties include a correction quantifying this discrepancy obtained by statistically comparing the original uncertainties to the measured scatter relative to the TCCON. As a consequence, the reported uncertainties of TROPOMI/WFMD are considered realistic. For the other data we used the $1\sigma$ uncertainties provided for each dataset. We refer to the documentations of the datasets for an explanation on how these uncertainties are calculated.

**Referee comment: Validation of results: The authors compare their results to several other annual methane increases (AMIs) obtained using different data and the same method or the same data and different methods. The comparison here is largely qualitative and doesn't attempt to explain the sources of the observed differences. It would be useful to understand what attributes of the different datasets or methods lead to the observed similarities or differences.**
Author's reply: See our answer regarding L. 253–254 below.

**Referee comment: Writing style**
Author's reply: In the updated manuscript we divided long sections into multiple paragraphs, rephrased some parts and added explanations to all equations.

**2.2 Specific comments**

**Referee comment: Line 80: I didn't understand what "model data" meant.**
Author's reply: We omitted "model" since the source of the data, the CAMS $CH_4$ Flux inversion system, is mentioned in the next sentence.

**Referee comment: Line 98: I was confused by the reference to satellite data. Which instruments are used?**
Author's reply: The Obs4MIPS $XCH_4$ data is based on data from the satellite instruments SCIAMACHY/ENVISAT, TANSO-FTS/GOSAT and TANSO-FTS-2/GOSAT-2. We added this to the paragraph.

**Referee comment: Lines 97 − 100: I was a bit confused by this data description. You refer to the "time series of monthly values of the column-averaged mole fraction of atmospheric methane," which seems to be at an almost-global spatial resolution between 60°S and 60°N. Is that correct? Later in the paper (Line 367) you refer to total-column data from UB-C3S-CAMS and seem to apply the DLM to this data. If indeed you use this raw data, it would be helpful to provide more details (on coverage, potential biases, etc., as described above for TROPOMI).**
Author's reply: Concerning the use of total-column data from UB-C3S-CAMS: It is not exactly clear what is meant with "raw data" but independent of that, we have not used "raw data". Which data we have used is described in detail in Sect. 2.4. In short, we have used a time series of monthly XCH4 for 60°S and 60°N (one XCH4 value per month, not spatially resolved). As described in Sect. 2.4, this is exactly the input data set also used for the computation of annual mean methane growth rates as shown in the corresponding climate indicator section of the Copernicus European State of the Climate 2022 assessment report in our manuscript). We updated the following sentence in the manuscript:
The $XCH_4$ time series corresponds to **averaged** satellite data over land in the latitude band 60° S – 60° N and covers the period January 2003 to December 2022.

**Referee comment: Line 106: It would be helpful to provide a brief summary of the approach used by Buchwitz et al. (2017).**
Author's reply: Buchwitz et al. (2017) describes a method to compute annual mean growth rates of satellite-derived XCO2. A later publication (Reuter et al. (2020)) describes an update of the $XCO_2$ result and first results for satellite-derived $XCH_4$. Here we use $XCH_4$ to shortly describe the method (for details please see Buchwitz et al. (2017)). The method is based using monthly averaged XCH4 for a pre-defined latitude band, e.g., 60°S and 60°N, and using only data over land (to minimize effects due to sparse ocean coverage of the underlying spatially resolved XCH4 input data set). From this monthly time series, a new time series is generated by computing the difference of the XCH4 value of a given calendar month (e.g., January 2020) and the corresponding value of the same month but the year before (e.g., January 2019). The time assigned to this difference is the time difference between the two months (e.g., mid-July 2019). The annual mean growth rate for a give year is the weighted average

of all monthly difference values of that year. Ideally, one could have simply computed the difference of XCH4 at the end of a year minus the XCH4 at the beginning of that year, but this has not been done mainly because of data gaps. We updated the paragraph by adding this explanation.

**Referee comment: Line 116: How is uncertainty defined? Have the uncertainties been evaluated? For example, I know that the SRON-provided TROPOMI uncertainty is biased low because it only considers instrument errors, and excludes retrieval errors. I would be curious, for example, if the uncertainties used here are larger over snow- and ice-covered scenes, or at low albedos, as I would expect.**
Author's reply: As described above, the reported uncertainties include an inflation based on an evaluation with the actually measured scatter relative to the TCCON. They are thus considered realistic by definition. The reported uncertainties are indeed larger at low albedo / surface spectral reflectance, because this increases the noise to signal ratio of the measurements of the XCH4.

**Referee comment: Lines 119-120: Please briefly define (intuitively) the asymmetry and entropy of the data. What metric do you use for asymmetry? Skewness?**
Author's reply: The asymmetry is defined as the (normalized) absolute difference between the mean location of measurements in a region (e.g. a zonal band) and the central point (e.g. the 0°E meridian). A day with measurements exclusively in the eastern/western hemisphere would hence have an asymmetry greater than 0. A day with only two measurements which are symmetrically arranged around the central point would have an asymmetry of 0. This is why the entropy is used as a second measure of inhomogeneity. The entropy is defined for the same region subdivided in bins (in our case the grid cells) and is given by the classic normalized Shannon entropy. Perfectly homogeneous data (i.e. no missing data) has an entropy of 1. For each missing grid cell the entropy is lowered. This however doesn't capture the distribution of the data, hence the combination of asymmetry and entropy to gain a more meaningful homogeneity measure. Figure 3 in Sofieva et al. (2014) visualizes this with some example distributions. We added additional explanation to this section.

**Referee comment: Lines 123 - 126: Please briefly distinguish between the temporal inhomogeneity and spatial inhomogeneity.**
Author's reply: We made some changes to this section to make the difference more clear and added specifically the following sentence: The inhomogeneity can be calculated in the temporal domain (for each grid cell) and in the spatial domain (for each time step).

**Referee comment: Lines 123 – 129: Did you confirm that the inhomogeneity metric removes data over timesteps or regions with sparse data? I wasn't sure if your statements that HT "tends to be higher in cells with sparse data coverage" and that the HS filtering process "removes days with highly inhomogeneous coverage" provided this confirmation.**
Author's reply: Yes. Figure 1 shows the temporal and spatial inhomogeneity for global data. The temporal inhomogeneity is the same used for zonal data (if

you just look at the zonal subgrids). It can be clearly seen that this criterion removes grid cells with inhomogeneous coverage over the oceans and tropical rain forests. The spatial inhomogeneity is different for every investigated region. For global data only a few days are removed, as shown in Fig. 1. For zonal bands this can vary. For example in the high latitudes this criterion often filters days close to the boundary of the polar night, when data coverage decreases.

**Referee comment: Lines 124 − 125, Figure 1: I was surprised that there isn't higher temporal inhomogeneity at high latitudes, where I would expect there to be significant seasonal inhomogeneity due to the variation in the amount of sun over the course of the year. Am I misunderstanding something about your metric?**
Author's reply: It is true that the seasonal data gaps introduce some inhomogeneity due to the decrease in the entropy measure. However, the asymmetry measure is not affected by these seasonal gaps. Hence, the overall inhomogeneity is not as high as one might expect.

**Referee comment: Lines 138: I was confused to which time series you add the Gaussian noise, the CAMS/INV data or the WFMDv1.8 time series?**
Author's reply: We add the Gaussian noise to the CAMS/INV data, we clarified this in the manuscript.

**Referee comment: Line 171: I was confused about what you solve for using your Kalman Filter. Is it the variances of the Gaussian random walks?**
Author's reply: The variances of the Gaussian random walks are estimated using the MLE. The Kalman Filter estimates the unobserved states for each time step, using the model equations and the observations. The unobserved states correspond to the components we use to model the data (i.e. level, trend, seasonality, AR(1)).

**Referee comment: Lines 174 − 176: Can you provide more information about the parameters you used in your MLE? For example, how do you define the errors on the observations?**
Author's reply: The MLE is used to estimate the variances used for the Gaussian random walks and the parameter $\rho$ of the autoregressive component. Table A1 shows an overview of all DLM parameters and whether they are determined by MLE, we added a reference to this table in the main text.
The error on observations is estimated from the data by MLE and is called $\sigma^2_{irr}$. It is not possible to directly input measurement errors into the DLM. The errors are however indirectly included, since we use the uncertainties given by the WFMDv1.8 product during the initial gridding of the data.

**Referee comment: Line 187: I was confused by what you meant by "variance in the level and seasonality," which you use later for model evaluation. In particular, I'm confused how seeking low variance in the level wouldn't incentivize low growth rates. I'm probably misunderstanding something!**
Author's reply: The variance is the square of the standard deviation, which we

use as an estimate of the uncertainty. Favoring low variance in level & seasonality therefore favors models with lower uncertainty in the corresponding components. This not necessarily connected to the inherent variability in these components. We will clarify this in the text by substituting uncertainty for variance.

**Referee comment: Line 193: I didn't understand what you meant by "the average MSE across all five folds per DLM provides the AMSE." I'm assuming this is standard k-fold cross validation, but I'd appreciate additional clarity!**
Author's reply: Correct, we are using standard k-fold cross validation. For each model configuration we perform a DLM fit on all k-folds (k=5 in our case). For each of these fits we get the MSE which we average to gain an averaged MSE or AMSE.

**Referee comment: Lines 209 − 210: Please define all terms even if they seem obvious (as requested above). I was also a bit confused by the indexing. i seems to index across the models in equation (4) but across the time steps in equation (5). As a result, I also didn't understand how equation (5) quantifies the model uncertainty without indexing over the models.**
Author's reply: We apologize for the confusion and thank you for noticing this mistake. The index $i$ is the cross model index, while we omitted the time index in the original version. We updated this section and (a) added time index to both equation and (b) clarified that for AMIs we calculate one model uncertainty for each year while for growth rates we calculate only one averaged uncertainty.

**Referee comment: Lines 224 − 226: I didn't understand why the WFMDv1.8 mask wouldn't include the effect of polar nights. Or are these meant to separate out the effect of polar nights from the total effect of TROPOMI sampling, including polar night effects?**
Author's reply: Correct, the WFMDv1.8 includes the effect of polar nights. However, we wanted to separate the polar night effect from the total effect of TROPOMI sampling.

**Referee comment: Lines 228 − 220: Are you referring here to the S5P masks from the WFMDv1.8 data? Does it or does it not include the additional masking to simulate polar nights?**
Author's reply: Yes, we mean the S5P mask derived from the WFMDv1.8 data. This includes the polar night mask by definition, since no TROPOMI measurements are available at this time.

**Referee comment: Line 232: What does "remaining bias due to sampling" refer to? What was the original bias?**
Author's reply: As we described in the section, we use our "zonal first averaging" approach to reduce the potential sampling bias. With "remaining bias due to sampling" we refer to all remaining sampling bias when using this averaging approach. We clarified this in the revised version.

**Referee comment: Lines 241 − 244: It would be helpful to specify that this is for each zonal band individually (perhaps "we calculate a zonal error" in Line 241).**
Author's reply: We clarified this in the revised version of the manuscript.

**Referee comment: Line 245: I didn't understand what the word "visualizes" intends to convey here.**
Author's reply: We clarified this in the revised version of the manuscript.

**Referee comment: Lines 250 − 251: Perhaps I'm not familiar with the convention in the field, but I would have found units of ppb/yr to make it clearer that you're comparing slopes. (Even though you are expressing the increase for a specific year.) I think this is also what you do later in the manuscript (e.g., Line 312).**
Author's reply: We are not directly comparing slopes (or growth rates) at this point but AMIs which are defined as the difference in the methane level between two points in time. Another way to describe the AMIs would be to define them as the integral over the growth rates over one year. In our view the unit *ppb* is therefore more appropriate, which is also the convention used in existing literature.

**Referee comment: Lines 253 − 254: Per my point (2) above, could you provide a brief description of the methods used for each of these comparison data sets?**
Author's reply: We believe that this is not within the scope of this manuscript, and that the interested reader can find better information under the given references. Our aim was to show, that our method delivers comparable results as the already established methods, which we showed by replicating AMIs for different datasets using our method within $1\sigma$.

**Referee comment: Table 2: Why aren't $\sigma_{DLM}$ and $\sigma_{sampling(SZA)}$ given here? I assume I'm missing something about why these quantities aren't provided for the zonal results—it would be good to clarify somewhere in the text.**
Author's reply: We added $\sigma_{sampling(SZA)}$ in the revised manuscript. $\sigma_{DLM}$ is not included in the table since the uncertainty varies for each time steps and can be better seen in Fig. 7 which includes this uncertainty.

**Referee comment: Lines 260 − 261: Do you mean ". . . agree well with the DLM-based AMIs for the UB-C3S-CAMS and NOAA-GML datasets"? Or do you mean that you compare your DLM-based AMIs for each dataset to values derived with other methods for each dataset? Please clarify. Can you also provide more details on what it means to "agree well"? Do your results agree within error bars? More generally, to my point (2) above, I would be interested in how much of the variance between the results is due to different datasets vs. different methods.**
Author's reply: Correct, we mean a comparison of AMIs when using the same input data but different methods (our DLM approach vs. the NOAA/Buchwitz approach). In general, AMIs agree when the same data is used (either WFMDv1.8(daily),

WFMDv1.8(monthly), NOAA MBLR or UB-C3S-CAMS). With agreement, we mean agreement within $1\sigma$, which we clarified in the text. We also added a short list of possible reasons for differences between the AMIs derived from different datasets. A deeper investigation is however not within the scope of our work.

**Referee comment: Figure 6: It would be helpful to, in the caption, note that you use the format @ in your naming convention. Also, is there a method to the color scheme? If so, can you explain this in the caption? If not, it might be useful to group together datasets with the same symbol and methods with the same color.**
Author's reply: We updated the figure and caption to explain the naming convention and coloring scheme (which we updated to be more clear).

**Referee comment: Line 283: What do you mean by the "identification of zonal bands with anomalous methane increases using zonal AMIs"?**
Author's reply: We wanted to suggest that one could use the same concept of AMIs used for global data to investigate zonal data with the goal to identify zonal bands with anomalous methane increases. We tried to clarify this part of the text.

**Referee comment: Lines 284 − 289: I agree that it's preferable to look at zonal information on monthly time scales, before it's well mixed, but would mixing completely prevent us from getting zonal information on annual or seasonal time scales? Wouldn't column data give information about the surface (i.e., newer, less mixed methane), providing zonal information even on annual or seasonal time scales? If so, it'd be useful to be less absolute in your statements about the timescales on which zonal information can be obtained.**
Author's reply: It could be possible to obtain some zonal information on larger timescales if the corresponding signal is strong enough (in magnitude and duration). However, the usage of growth rates is hence preferable in our case and makes the best use of the available temporal resolution.

**Referee comment: Figure 7: In general, I'd use either -90° (no directional suffix) or 90°S. I was confused by the reference to -90°N. For consistency with the text, I'd use 90°S.**
Author's reply: We updated the figure to use 90°S style.

**Referee comment: Line 312: "For 2020 growth rates increase strongly from roughly 0 ppb/y to 20 ppb/y." Are these global growth rates or southern hemisphere growth rates? If the latter, are you averaging across all southern hemisphere bands?**
Author's reply: Here we were talking about all SH bands which show roughly the same behavior for this year. We clarified this in the text.

**Referee comment: Lines 315 − 316: The CAMS/INV-SRF results seem to show growth rates in the southern hemisphere remaining relatively constant for this year, compared to the decrease you see. Do you have thoughts on why this may be?**

Author's reply: We suspect that this could be related to the sparse ground measurement network in the SH (compared to the NH), used in the CAMS/INV-SRF data. This is supported by the fact that growth rates for CAMS/INV-SRF-SAT show a better agreement for the 30°-50°S band (see Fig. C2) with our results, which also features the same decrease of growth rate at the end of 2021.

**Referee comment: Line 324: Do you have evidence from your work to support this claim or are you relying on consistency with past studies? If the latter, it may be clearer to write that the increase is "consistent with increased southern hemisphere wetland emissions" might be better. (I agree! You find good support for this claim. But seeing an increase in southern hemisphere emissions alone does not necessarily imply increased wetland emissions.)**
Author's reply: We agree and used "consistent" in the updated manuscript.

**Referee comment: Lines 327 − 329: As above, do you have evidence from your own work to support this claim? In order to make the argument that it's attributable to "the return to pre-pandemic methane emissions from the energy sector," you would need to show that the northern hemisphere methane emissions decreased during the pandemic despite the increase in oil and gas emissions. This is beyond the scope of this work, though you could certainly cite other studies that show this and argue that your results are consistent with those results.**
Author's reply: We agree and removed this line from the manuscript.

**Referee comment: Lines 335 − 336: You write, "Comparison of our global AMIs with global AMIs from other sources indicate, [sic] that the effect of transport related sampling biases seems to be limited." This is interesting—can you justify it a bit more? Isn't it possible that there are consistent biases across the data sets or methods?**
Author's reply: Our argument is twofold. First we demonstrated that differences between methods are within $1\sigma$ and since the methods vary widely in their approach (whether they model the data explicitly or not, how sampling related biases are treated, etc.) we surmise that a consistent bias underlying all method is highly unlikely. This leaves the data which could share the same consistent biases. Here we argue that all data sets share their problem with incomplete sampling of the globe, but since their individual sampling varies to a large degree we would again surmise that a similar bias between all methods seems unlikely. The agreement between our AMIs and AMIs from NOAA (within 2-sigma) further strengthen this point because the fundamental difference in sampling (large amount of satellite data between 90°S-90°N vs. single ground stations across the globe). Additionally, we tested the effect of S5P sampling using CAMS/INV-SRF data and showed that no significant sampling bias can be observed when employing our zonal first averaging approach (see Figure 3 in the manuscript). We believe that the latter argument is the stronger and more fitting, so we changed this passage in the manuscript accordingly.

**Referee comment: Lines 338 − 340: I was a bit confused by this statement because the area where you find the largest differences between**

**WFMD and CAMS/INV is also where your inhomogeneity metric is the largest (the southern tropics). Can you rule out artifacts from sampling related biases?**

Author's reply: We want to highlight, that zonal growth rates agree within $1\sigma$ between WFMD and CAMS/INV. It is true that there is some disagreement in areas with challenging measurement conditions. However, this is reflected in the higher sampling bias (see Tab. 2), which is included in the errorbars of the WFMD growth rates. Since sampling is mostly uniform between the years, differences between WFMD and CAMS/INV are likely not related to sampling biases.

**Referee comment: Line 355: Instead of "other fluxes," it might be clearer to write "non-wetland fluxes." This is a personal preference, though.**

Author's reply: We agree and changed this in the revised manuscript.

**Referee comment: Lines 356 − 357: This again feels a bit beyond what you are able to show in this work. Might it be better to say "suggests" instead of "indicates"?**

Author's reply: We agree and changed the wording in the revised manuscript.

**Referee comment: Line 365: Is this true? From visual inspection of Figure 6, it seems that there are a number of studies for which the error bars don't overlap (e.g., NOAA-GML (v2023-05)@NOAA MBLR for 2019 and 2022).**

Author's reply: This is true, what we meant is that using the same input data but different method results in agreements within $1\sigma$. Results derived using the same method but different data show qualitative agreement but larger absolute differences. We updated the section accordingly.

**Referee comment: Line 373: Is it true that "no significant sampling biases exist for zonal bands"? See my previous comments (on Lines 338 − 340) about potential bias in the southern tropics.**

Author's reply: We extended Section 3.4. to include a comparison of growth rates derived from CAMS/INV data using either no mask or the S5P $XCH_4$ mask. The (new) Figure 4 demonstrates that the agreement between both growth rates is good and therefore strengthens the point that no sampling bias exists for zonal bands.

**Referee comment: Lines 384 − 385: You made a different argument in Section 6 as to why transport changes aren't causing the observed growth rates. I find this one less convincing: an inverse system could easily alias transport errors into fluxes—this is a significant source of error in inverse modeling.**

Author's reply: We removed this sentence and reformulated this part.

**Referee comment: Lines 387: As above, I'm not sure your study allows you to draw firm conclusions on the causes of the changes in the growth rates.**

Author's reply: We change this part of the manuscript to make it more clear

that we cannot conclusively explain the changes in growth rates, but that the fluxes provide some possible explanation.

**Referee comment: Lines 390 - 391: The low computational cost, support for low-latency updates, and lack of reliance on a prior are neat features of this method! It might be worth mentioning sooner!**
Author's reply: Thank you for this comment, we now mention these features of our approach in our abstract.

**Referee comment: Line 396: You write, "This indicates. . . ." What is "this"?**
Author's reply: With this we refer to the observation made in the previous sentence.

**3 Reviewer 2**

**3.1 General comments**

**Referee comment: The authors apply DLM to a few datasets, some are concentrations and others are emissions. It is not clear to me how these datasets are treated differently in the analyses. For instance, the increase in concentration should be affected by both emissions and methane losses, whereas CAMS/INV data should only reflect changes in emissions. This should be clearly stated in the result discussion and comparison.**

Author's reply: We only apply our DLM method to concentrations, either to satellite data, to CAMS/INV model data (both given as dry column mixing ratios) or to NOAA–GML mixing ratios of CH4 at the surface. Additionally, we use the corresponding emission fluxes for the CAMS/INV model data, to investigate whether changes in growth rates correspond to changes in the fluxes (and are not just result of transport related effects). We clarified in the corresponding sections that we apply our DLM to concentrations from the CAMS/INV model data, by explicitly referring to "CAMS/INV XCH$_4$ data".

**Referee comment: Section 3.2 has a mixture of literature review and method. It is not clear what exactly has been applied in the DLM fit in this work based on the descriptions. For instance, the authors mentioned the Kalman Filter in line 172. Has it been applied to this work? If so, what are the detailed setups? If not, this information should be removed to avoid confusion. The rest of this section has similar issues.**

Author's reply: We divided this section into multiple paragraphs to make it more comprehensible. We don't implement the Kalman Filter ourselves, but it is used by the UnobservedComponents class in combination with Maximum Likelihood Estimation to estimate the states and parameters. To avoid confusion we removed mention of the Kalman Filter from the text and refer to the documentation of the respective software package.

**Referee comment: In addition, the analyses of the cause of methane increase are very vague. Figure 10 and Figure 11 separately show emission changes from wetlands and other sources, but how are they separated? Why is it only presented for this dataset? In lines 354-355, it is not clear how the authors did the source attribution, or these are just hypotheses. In most other places, the authors only cite previous work to explain the sources of methane changes. This gives me the impression that this work does not add much new to explain the methane increase signals.**

Author's reply: Figure 10 and 11 show the wetland/non-wetland fluxes for the CAMS/INV data. We do not undertake the surface flux inversions at the University of Bremen. We use the fluxes provided by the CAMS/INV system. The surface fluxes are therefore only presented for the model data. However, we use them to understand why the growth rates for the model data are changing. Our reasoning in line 354-355 was, to explain the growth rates derived from satellite data by understanding the growth rates derived from model data (which

show a very similar structure). Since emission fluxes are readily available for the model data we can use these to investigate the changes in fluxes between the years. We can therefore show, by using the comparison to model data, that our zonal growth rates derived from satellite data are consistent with existing literature. Based on these results, which provide validation of our method, we draw the new conclusion that (a) the reduction of global AMI in 2022 is caused by the reduction of NH growth rates and (b) Changes in NH growth rates are clearly correlated to changes in NH fluxes. It is true that this is an inferred connection and needs further research (which is however not in the scope of this manuscript). We updated this part of the manuscript to make the distinction between the reproduction of existing results and our new conclusions more clear.

---

## Author Response (AR2)

**Author reply**

Jonas Hachmeister

01.12.2023

**1   Answer to the editor**

We would like to thank the editor and the reviewer for their help with improving the manuscript.

In the following we list the reviewer comments in **bold** font and our answers in normal font.

**2   Answer to Reviewer 1**

**2.1   Specific comments**

**Referee comment: Lines 15 − 16 and 72 − 73: It would help to specify that the CAMS global inversion system uses surface observations and data from GOSAT.**
Author's reply: We mainly compare our results to growth rates derived from CAMS/INV data based only on surface observations. To keep the abstract short we hence only mention the CAMS/INV data based on surface observations. In the introduction we mention briefly of both versions of CAMS/INV (with and without satellite data).

**Referee comment: Lines 21 − 22: Consistent with the second reviewer's last comment, it might help to rephrase this sentence to clarify that the author's interesting results support past conclusions about the causes of observed methane trends rather than producing new inferences.**
Author's reply: We added a sentence to the abstract to mention that our results support past conclusions about the causes of observed methane trends. However, we still want to emphasize that the discrepancy between NH and SH methane increases is a novel result of our study and, to our knowledge, hasn't been mentioned before in literature.

**Referee comment: Line 116: In the response to reviewers, the authors seem to suggest that the data are averaged in two latitude bands at 60°S and 60°N, instead of one latitude band between 60°S and 60°N as suggested here. I'm assuming this was a typo, but the proper explanation should be used here.**
Author's reply: We apologize for the misunderstanding in our reply. Indeed we

mean a single region between 60°S and 60°N as described in the manuscript.

**Referee comment: Line 141: Many thanks to the authors for their description of how the uncertainties are calculated in the response to the reviewers! Can you mention the origin of the uncertainties used here somewhere in the text (perhaps in the section 2 data descriptions?).**
Author's reply: As recommended we describe the origin of uncertainties to section 2.

**Referee comment: Line 286 − 287: Thanks so much for clarifying these terms! I assume the standard deviations here to the DLM-produced uncertainties? (Can you clarify in the text, too (e.g., as "the corresponding uncertainties as given by the DLM")?)**
Author's reply: As recommended we clarified this in the updated manuscript.

**Referee comment: Figure 4 and lines 330 − 337: I'm confused by this figure. What does it mean to apply zonal-first averaging to zonal growth rates? You describe zonal-first averaging as calculating the zonal average, then averaging the zonal averages globally. I assume you only apply the first step here. As a result, I'm confused by what you show in Figure 4. I'm also confused by the large differences between zonal-first averaging and full-coverage averaging at 10°S − 10°N, which seem to suggest that the standard averaging approach may be preferable. It also seems like neither averaging method recovers the full-coverage growth rate at 30°S − 50°S.**
Author's reply: By zonal-first averaging we mean that the data is first averaged in each latitude "band" and then averaging all latitude band averages. Since the data is on a 2°x2° grid this means that each latitude band is a 2° band. This is what we mean by zonal bands. We understand that this was not explained sufficiently and provided some clarification in the updated manuscript. Zonal-averaging first for zonal bands hence means that (for a 20° zonal band), we first average the 2° latitude bands and then calculate the average of the 20° band from these ten averages. Zonal-averaging first can thus also be applied to 20° zonal bands.
Regarding the 10°S-10°N band we want to point out that both averaging methods are in $1\sigma$ agreement with the growth rate derived from unmasked data. It is also true that neither method recovers the full-coverage growth rate at 30°S-50°S, indicating that sampling is to sparse to recover the full information. This is why we calculate a sampling bias in the following paragraph which quantifies the difference between growth rates derived from unmasked and masked data. For the aforementioned regions this sampling uncertainty is especially high (see Tab. 1). Both averaging methods show good agreement for most bands, and some disagreement for few bands, with no clear winner. As mentioned in the manuscript, we therefore base our decision for using zonal-first averaging mainly on the fact that sampling within a 20° zonal band can still vary with latitude and zonal-first averaging is hence preferable.

**Referee comment: Line 380: Is this related to the higher uncertainty for AMIs at the start/end of a time series or the higher uncertainty**

for DLMs at the start/end of a timeseries? If the former, why does this explain the deviation of DLM@CAMS/INV-SURF compared to the other AMIs shown in Figure 7?

Author's reply: The latter is correct. The higher uncertainty for AMIs at the start/end of a time series is directly caused by the higher uncertainty for DLMs at the start/end of a time series. We changed this in the text.

**Referee comment: Line 424: Are there no significant sub-annual variations in zonal growth rates? Figure 8 panels 30°N − 50°N, 50°N − 70°N, and 50°S − 70°S seem to suggest otherwise. (Relatedly, on line 426, you write "short-term variations between zonal bands are not detected.")**

Author's reply: By sub-annual variations we actually meant variations on a monthly timescale. We clarified this in the updated manuscript.

**Referee comment: Line 431: Do you mean "bands between 10°N (instead of S) and 70°N when we speak of the Northern Hemisphere"? It seems like you discuss 10°S − 10°N separately.**

Author's reply: We corrected this typo in the updated manuscript.